# The laminar organization of the *Drosophila* ellipsoid body is semaphorin-dependent and prevents the formation of ectopic synaptic connections

Xiaojun Xie[1,2], Masashi Tabuchi[3], Matthew P Brown[1,2], Sarah P Mitchell[1,2], Mark N Wu[1,3], Alex L Kolodkin[1,2]*

[1]The Solomon H. Snyder Department of Neuroscience, The Johns Hopkins University School of Medicine, Baltimore, United States; [2]Howard Hughes Medical Institute, The Johns Hopkins University School of Medicine, Baltimore, United States; [3]Department of Neurology, The Johns Hopkins University School of Medicine, Baltimore, United States

*For correspondence: kolodkin@jhmi.edu

Competing interests: The authors declare that no competing interests exist.

**Abstract** The ellipsoid body (EB) in the *Drosophila* brain is a central complex (CX) substructure that harbors circumferentially laminated ring (R) neuron axons and mediates multifaceted sensory integration and motor coordination functions. However, what regulates R axon lamination and how lamination affects R neuron function remain unknown. We show here that the EB is sequentially innervated by small-field and large-field neurons and that early developing EB neurons play an important regulatory role in EB laminae formation. The transmembrane proteins semaphorin-1a (Sema-1a) and plexin A function together to regulate R axon lamination. R neurons recruit both GABA and GABA-A receptors to their axon terminals in the EB, and optogenetic stimulation coupled with electrophysiological recordings show that Sema-1a-dependent R axon lamination is required for preventing the spread of synaptic inhibition between adjacent EB lamina. These results provide direct evidence that EB lamination is critical for local pre-synaptic inhibitory circuit organization.

## Introduction

Proper nervous system function relies on precise synaptic connectivity. Laminated distribution of synaptic connections is an organizational feature observed in both vertebrate and invertebrate nervous systems (*Kolodkin and Hiesinger, 2017*; *Baier, 2013*; *Sanes and Yamagata, 2009*). However, the cellular and molecular mechanisms that control neuronal lamination remain to be fully elucidated, as does how laminar constraint of synaptic organization in particular brain structures contributes to function.

The insect brain, although small, is composed of myriad complex synaptic connections organized into multiple neuropil modules (*Ito et al., 2014*), and each module adopts distinct structural features geared toward specific functions. For example, in *Drosophila* the optic lobes include heavily laminated synaptic connections (*Sanes and Zipursky, 2010*), while the antennal lobes include individual glomeruli where distinct olfactory sensory neurons contact to second-order neurons (*Wilson, 2013*). Deep within the insect brain, a highly conserved neuropil module called the central complex (CX) is composed of several laminated structures critical for sensory integration and motor coordination functions analogous to the vertebrate basal ganglion (*Strausfeld and Hirth, 2013*; *Strausfeld, 2012*; *Turner-Evans and Jayaraman, 2016*).

**eLife digest** The human brain contains around one hundred billion nerve cells, or neurons, which are interconnected and organized into distinct layers within different brain regions. Electrical impulses pass along a cable-like part of each neuron, known as the axon, to reach other neurons in different layers of various brain structures. The brain of a fruit fly contains fewer neurons – about 100 thousand in total – but it still establishes precise connections among neurons in different brain layers.

In both flies and humans, axons grow along set paths to reach their targets by following guidance cues. Many of these cues are conserved between insects and mammals, including proteins belonging to the semaphorin family. These proteins work together to steer growing axons towards their proper targets and repel them away from the incorrect ones. However, how neurons establish connections in specific layers remains poorly understood.

In the middle of the fruit fly brain lies a donut-shaped structure called the ellipsoid body, which the fly needs to navigate the world around it. The ellipsoid body contains a group of neurons that extend their axons to form multiple concentric rings. Xie et al. have now asked how the different "ring neurons" are organized in the ellipsoid body and how this sort of organization affects the connections between the neurons. Imaging techniques were used to visualize the layered organization of different ring neurons and to track their growing axons. Further work showed that this organization depends on semaphorin signaling, because when this pathway was disrupted, the layered pattern did not develop properly. This in turn, caused the axons of the ring neuron to wander out of their correct concentric ring and connect with the wrong targets in adjacent rings.

Together these findings show that neurons rely on evolutionarily conserved semaphorins to correctly organize themselves into layers and connect with the appropriate targets. Further work is now needed to identify additional proteins that are critical for fly brains to form layered structures, and to understand how this layered organization influences how an animal behaves.

The CX is composed of four neuropil substructures, from anterior to posterior: the ellipsoid body (EB), the fan-shaped body (FB), the noduli (NO) and the protocerebral bridge (PB) (*Figure 1A*). In each CX substructure, neuronal processes elaborate their trajectories in precisely defined regions. They form multiple layers, or laminae, from dorsal-to-ventral within the EB, FB and NO, and other neuronal processes form 16–18 columns, medial-to-lateral, within the EB, PB and FB. More than 50 different types of 'small-field' and 'large-field' neurons innervate these CX substructures (*Hanesch et al., 1989*; *Young and Armstrong, 2010b*). Every small-field neuron innervates one or two columns and contacts one or multiple laminae in specific CX substructures. On the other hand, each large-field neuron innervates an entire single lamina across all columns in certain CX substructures. Therefore, small- and large-field neurons collaboratively form highly organized wiring patterns and are interconnected in all four CX neuropils (*Lin et al., 2013*; *Wolff et al., 2015*; *Yang et al., 2013*), allowing for information flow in precisely defined patterns within the CX.

The EB in *Drosophila* is the anterior-most CX substructure and adopts a toroid shape in the central brain (*Figure 1A*). The major large-field neurons that innervate the EB are called ring (R) neurons, named for their circumferential ring-like axonal arborization patterns that form several circular laminae/rings in the anterior shell of the EB (*Hanesch et al., 1989*). Anatomical analyses suggest that ring neurons, through their dendrites located in the bulb (*Figure 1B*, arrows), receive pre-synaptic inputs from descending neurons of the anterior visual pathway (*Omoto et al., 2017*). Indeed, neuronal activity of some ring neuron dendrites correlates with light stimulation and is organized in a retinotopic fashion within the bulb (*Seelig and Jayaraman, 2013*). Further, some ring neurons contribute to the circuits that influence visually evoked memory and learning (*Wang et al., 2008*; *Pan et al., 2009*; *Neuser et al., 2008*; *Ofstad et al., 2011*), and they are also involved in homeostatic regulation of behaviors such as hunger sensation (*Park et al., 2016*; *Dus et al., 2013*) and sleep homeostasis (*Liu et al., 2016*). One type of small-field neuron, called 'pb-eb-gall' or 'E-PG,' elaborates dendrites in the EB that overlap with R neuron axons and presumably serves as their post-synaptic partners. Neuronal activity observed in pb-eb-gall neuron dendrites is tuned to the

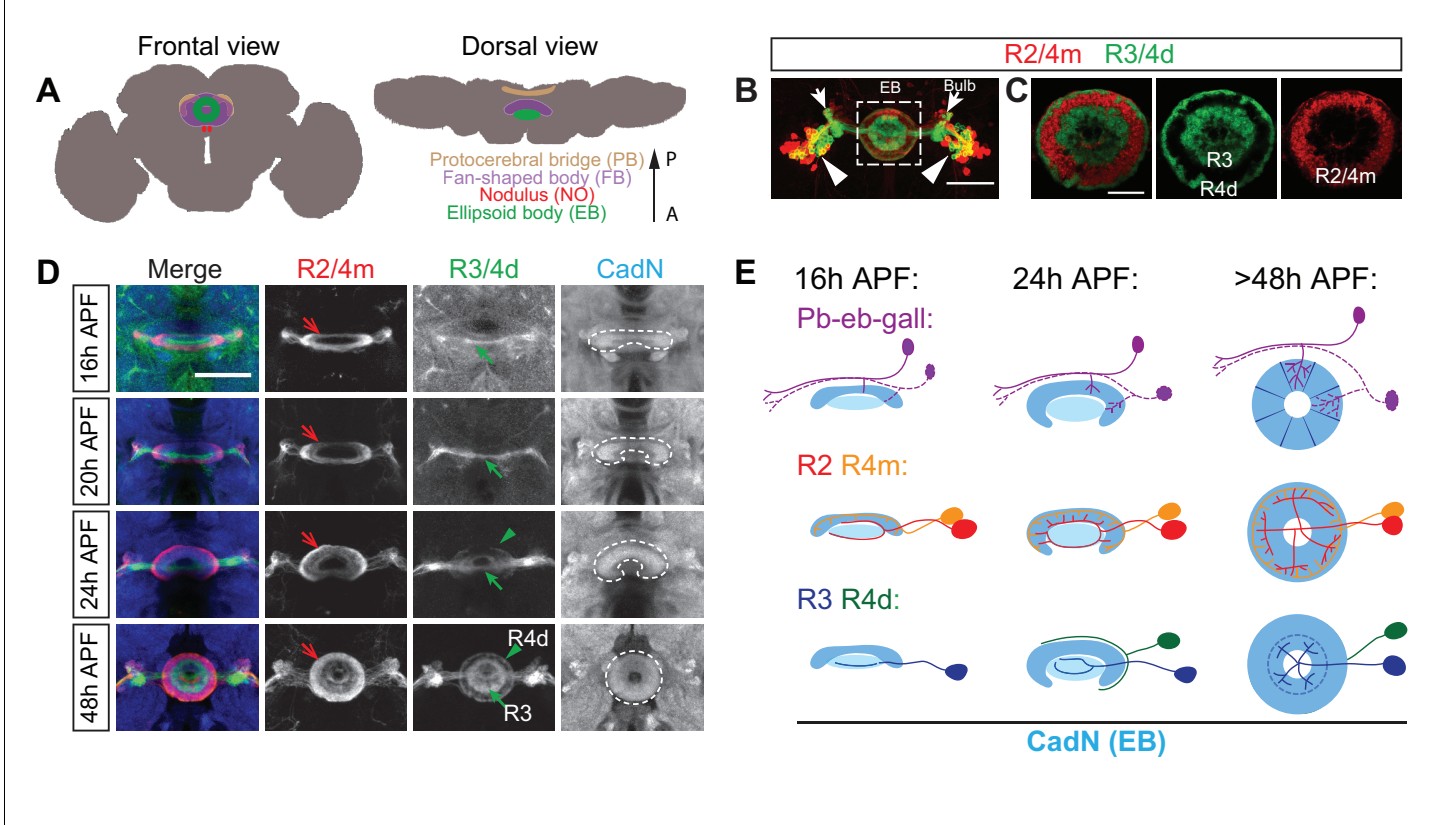

**Figure 1.** Large-field ring neuron axons sequentially innervate the ellipsoid body. (**A**) Schematics showing frontal and dorsal views of the central complex (CX) in an adult *Drosophila* brain. The CX is composed of four substructures: from anterior ('A') to posterior ('P') the ellipsoid body (EB), the noduli (NO), the fan-shaped body (FB) and the protocerebral bridge (PB). (**B–C**) *R15F02-GAL4* (expressed in R1/R3/R4d neurons) and *R32H08-lexA* (in R2/R4m neurons) driving GFP and mCherry reporters, respectively, label multiple ring neuron types in an adult fly brain. From laterally located cell bodies (arrowheads), ring neurons axons follow similar trajectories to innervate the bulb (arrows) and the ellipsoid body (dashed square) (**B**). Inside the EB, multiple concentric and adjacent rings are formed by dense axon projections from different R neurons (**C**). A smaller R1 ring is formed at more posterior EB regions and is covered by the R3 ring. (**D**) *R32H08-lexA*-driving mCherry and *R15F02-GAL4*-driving CD8-GFP allow for visualization of how R2/R4m and R3/R4d innervation takes place in the EB during pupal development. Strong CadN immunolabeling shows EB morphological changes (dashed lines) between 16 hr and 48 hr APF (hours after puparium formation). R2/R4m axons (red arrows) extend into the developing EB earlier than R3/R4d axons (green arrows and arrowheads), as can be observed between 16 and 24 hr APF since R2/R4m axons exhibit dense elaboration and co-localization with strong CadN staining prior to R3/R4d axons. R axon projections and elaborations appear complete between 40 and 48 hr APF, and the overall organization of R axons after 48 hr APF shows no difference when compared to the adult brain. (**E**) Schematics highlight changes in EB morphology (strong CadN staining in dark blue and weak CadN staining in light blue), sequential innervation of the EB by pb-eb-gall dendrites (immunostaining in *Figure 1—figure supplement 1K and L*), R2/R4m and R3/R4 axons during early pupal stages. Scale bars are 50 μm in panels **B** and **D**; 20 μm in panel **C**.

The following figure supplement is available for figure 1:

**Figure supplement 1.** R neuron axons and pb-eb-gall dendrites have different morphologies and closely associate with each other in the developing and adult brain.

fly's head direction coordinates (*Seelig and Jayaraman, 2015*; *Turner-Evans et al., 2017*; *Green et al., 2017*). How laminar organization of R axons influences R and pb-eb-gall neuron functions, however, remains unknown.

To gain insight into the logic of EB neural connectivity, we investigate here the developmental principles and molecular mechanisms underlying the formation of R neuron circumferential lamination and pb-eb-gall dendrite targeting within the EB. We observe sequential innervation of the developing EB by pb-eb-gall neuron dendrites and distinct R neuron axons during pupal

development, and we uncover different roles for pb-eb-gall dendrites and R axons in EB patterning. The transmembrane semaphorin semaphorin-1a (Sema-1a) is expressed in multiple types of R neurons and functions with the transmembrane protein plexin A (PlexA) to regulate ring neuron laminar organization in the EB. Disruption of Sema-1a signaling in the EB provides anatomical and functional insight into establishment of local inhibitory connections among R neurons that target adjacent EB laminae. These results show that local axonal interactions, through short-range repellent cues and receptors, regulate the assembly and functional organization of laminated pre-synaptic inhibitory circuits in the central complex within the fly brain.

## Results

### Ring neuron axons sequentially innervate the ellipsoid body during early pupal stages

All ring neurons share a common progenitor lineage and follow similar trajectories as their axons approach the brain midline (*Yang et al., 2013*). Multiple R neuron types have been identified based on their different axonal arbor morphologies in the EB and also by analysis of different enhancer trap lines that uniquely express the transcriptional activator GAL4 in R neuron types (*Hanesch et al., 1989*; *Renn et al., 1999*). For example, R1, R2 and R3 neurons have aster-like axon arbors (*Figure 1—figure supplement 1, A–C'*), and R4m and R4d neurons develop ring-shaped axon arbors (*Figure 1—figure supplement 1, D–E'*). Axon arbors from different R types also have different diameters and synapse distribution patterns within the EB. Together, they form multiple concentric circumferential laminae/rings in the EB (*Figure 1B and C*).

N-Cadherin (CadN) is a cell adhesion molecule widely expressed in the developing nervous system that consolidates target selection between pre- and post-synaptic neurons (*Iwai et al., 1997*; *Lee et al., 2001*). CadN immunostaining shows that the ellipsoid body forms during early pupal stages (*Young and Armstrong, 2010a*). Our results confirm that the EB emerges at around 16 hr (16 hr) after pupae formation (APF) with an open crescent shape that in later developmental stages closes toward the ventral side; the complete circular EB morphology is apparent between 40 and 48 hr APF (*Figure 1D*, white dashed lines in panels on right; *Figure 1E*, light blue schematics).

To track EB neuron innervation while overall EB morphology develops, we used both GAL4 and lexA drivers to label processes from multiple types of EB neurons simultaneously (*Jenett et al., 2012*; *Pfeiffer et al., 2008*). *R32H08-lexA* and *R15F02-GAL4* were first utilized to express mCherry in R2/R4m neurons and CD8-GFP in R3/R4 neurons, respectively, allowing us to trace ring neuron axon extension during early pupal stages (*Figure 1D*, red and green arrows).

At 16 hr APF, the earliest time point when it is possible to identify the EB by CadN expression, many R2/R4m axons have already reached the EB. These R2/R4m axons form a circular structure and the dorsal region of this axon trajectory overlaps with strong CadN staining, suggesting that R2/R4m axons have already established specific contacts with other neuronal processes at this early pupal stage. In contrast, CD8-GFP-labeled R3 axons were barely observed at the midline at 16 and 20 hr APF, and they appeared to have simple linear morphologies (*Figure 1D*, green arrows). These R3 axons are located in the central canal region of the developing EB defined by very weak CadN staining, suggesting that few CadN-mediated interactions have been established among R3 axons and other neurons at these early stages. At 20–24 hr APF, R2/R4m and R3 axons continue to innervate the developing EB, and R4d axons reach the edge of the EB and circle around the contralateral portion of the EB (*Figure 1D*, green arrowheads). All R neuron axons continue to extend between 24 and 48 hr APF, forming multiple rings with densely packed R neuron axons.

These observations show that the EB is sequentially innervated by axons from different R neuron types. R2/R4m neurons precede R3 neurons, and R4d neuron axons follow the axons of these other R neuron types. Moreover, R2/R4m, R3 and R4d axons do not overlap with one another during all the pupal stages we assessed, suggesting that they are segregated from each other soon after they arrive at the EB.

## Pb-eb-gall neuron dendrites sequentially associate with R2/R4m and R3 axons during ellipsoid body formation

In contrast to R neuron axons, small-field pb-eb-gall neurons have columnar dendritic elaborations in the EB, extending across multiple rings over the entire EB radius (*Figure 1—figure supplement 1F and H*). When we used the GFP-reconstitution-across-synaptic-partners (GRASP) technique to determine the proximity of pb-eb-gall neuron dendrites to R neuron axons, functional GFP was reconstituted between pb-eb-gall dendrites and R2/R4m (or R1/R3/R4d) axons in adult fly brains (*Figure 1—figure supplement 1I and J*, green fluorescence), revealing that un-laminated pb-eb-gall dendrites closely associate with multiple types of laminated R axons (*Figure 1—figure supplement 1G*).

To further characterize the relationship between pb-eb-gall dendrite and R neuron axon development, we used a combination of GAL4 and lexA drivers to label these neurons. Using the *R19G02-GAL4* driver line (*Wolff et al., 2015*), we found that many pb-eb-gall neurons, including their PB and Gall projections, are already present at the onset of the metamorphosis (data not shown). However, pb-eb-gall dendrites in the EB appear to develop after the onset of metamorphosis. At 16–24 hr APF, pb-eb-gall dendrites adopt a crescent shape within the EB (*Figure 1—figure supplement 1K and L*, blue dashed lines). They overlap with R2/R4m axons (*Figure 1—figure supplement 1K*, red arrows) but do not overlap with R3/R4d axons at these stages (*Figure 1—figure supplement 1L*, red arrow). At later pupal stages, pb-eb-gall dendrites extend ventrally to form a complete circle and also expand inwardly such that pb-eb-gall dendrites occupy the entire radius of the EB and overlap with both R2/R4m and R3/R4d axons, as can be appreciated at 48 hr APF (*Figure 1—figure supplement 1K and L*, bottom panels).

These data suggest that the development of pb-eb-gall dendrites and R axons is highly correlated during the EB formation (*Figure 1E*). Further, the select association between pb-eb-gall dendrites and R2/R4m axons at early pupal developmental stages suggests an instructive role for pb-eb-gall neurons with respect to R neuron axon targeting and lamina formation.

## Small-field and large-field neurons serve different roles in EB patterning

To address whether R neuron and pb-eb-gall neurons rely upon each other for the establishment of their axonal and dendritic elaboration patterns, we conducted a series of cell ablation experiments using specific GAL4 drivers to express a diphtheria toxin subunit (*UAS-DTI*) in select EB neuron types. A temperature-sensitive *tub-GAL80ts* was used to control expression of *UAS-DTI* and other UAS transgenes in embryos and larvae, and also to minimize non-specific effects from ablating early developing neurons (*Figure 2A*).

First, to ask whether small-field neuron dendrites influence R neuron axon patterning, we ablated a subset of pb-eb-gall small-field neurons by using the *R19G02-GAL4* driver to express CD8-GFP and DTI. At 24 hr and 48 hr after temperature shift (ATS) (comparable to 24 hr and 48 hr APF under normal animal rearing conditions), GFP-labeled pb-eb-gall cell bodies and processes were greatly reduced in number, and the remaining neurons exhibited much weaker GFP expression in DTI-expressing animals compared to control animals (*Figure 2B* and *Figure 2—figure supplement 1A*, compare arrows and arrowheads). This demonstrates successful DTI-mediated pb-eb-gall small-field neuron ablation.

Following loss of pb-eb-gall neurons, the EB developed major aberrant morphologies, including irregular edges and a loss of the EB central canal (data not shown). Further, R2/R4m neurons, labeled by using *R32H08-lexA*-driving myristoylated tdTomato (mtdT), no longer extended axons to form coherent ring-shaped structures in the DTI-expressing animals 48 hr ATS (*Figure 2B*, red circles and ovals). Similar lamination defects were observed in R3/R4d axons, labeled by *R70B04-lexA*-driving mtdT, when pb-eb-gall neurons were ablated (*Figure 2—figure supplement 1B*, red circle and oval). Both R2/R4m and R3/R4d axons appear to occupy the entire EB and therefore likely intermingle with one another in DTI-expressing animals. However, ablation of another type of small-field neuron called 'pb-eb-no' or 'P-EN' (*Wolff et al., 2015*; *Omoto et al., 2017*; *Turner-Evans et al., 2017*) that also elaborates processes in the EB during early pupal stages, using the *R83H12-GAL4* driver, did not cause apparent defects to the R2/R4m ring morphologies (data not shown). These results show that pb-eb-gall neurons are selectively required for instructing R neuron axon patterning during EB development.

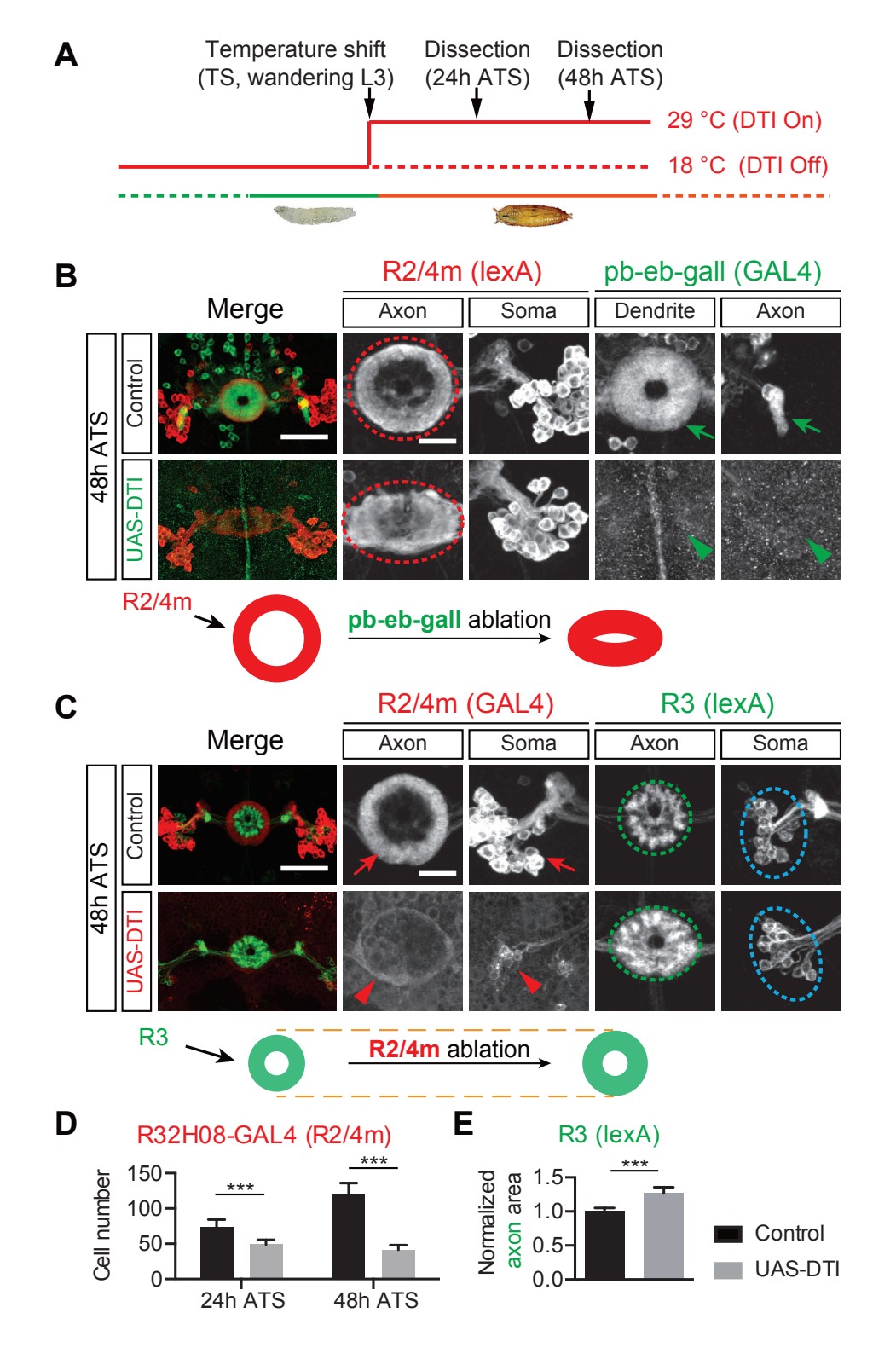

**Figure 2.** The pb-eb-gall and R2/R4m axons are required for EB patterning. (**A**) Genetically encoded diphtheria toxin (DTI) was conditionally expressed by GAL4 drivers in targeted neurons upon a temperature shift (TS) from 18°C to 29°C, starting from wandering third instar stage (wL3). Pupae were dissected 24 or 48 hr after the temperature shift (ATS). (**B**) The pb-eb-gall neurons were labeled by *R19G02-GAL4*-driving CD8-GFP (green). R2/R4m neurons were labeled by *R32H08-lexA*-driving mtdT (red). In control animals (n = 8), pb-eb-gall neuron dendrites and R neuron axons (red circle) form

*Figure 2 continued on next page*

*Figure 2 continued*

circumferential innervation at 48 hr ATS. DTI expression by *R19G02-GAL4* from wL3 stages ablated most pb-eb-gall neurons (arrowheads) and led to pronounced R2/R4m axon patterning defects in the EB (red oval) (n = 14 animals). (**C–E**) *R54B05-lexA* was used to drive myristoylated tdTomato (mtdT) in R3 neurons (green) while *R32H08-GAL4*-driving CD8-GFP was used for labeling R2/R4m neurons (red). In panel **D**, the CD8-GFP-labeled R2/R4m neuron cell bodies were counted at 24 hr ATS (73.4 ± 10.9 cells (n = 8 brains) for the control group and 49.0 ± 6.4 cells (n = 9 brains) for DTI expression group, p<0.0001) and 48 hr ATS (121.1 ± 15.0 cells (n = 7 brains) for the control group and 41.4 ± 6.9 cells (n = 10 brains) for DTI expression group, p<0.0001). In control animals, R2/R4m and R3 axons projected into different EB rings at 48 hr ATS (top panels in **C**). After DTI was expressed in R2/R4m neurons from wL3 stages, R3 axon arbors expanded outward in the EB (green oval in **C**). R3 axon arbor areas were measured and all measurements were normalized to the mean value of the control group shown in panel **E**: 1.000 ± 0.052 for controls (n = 8 brains) and 1.266 ± 0.090 for DTI-expressing group (n = 7 brains), p=0.0003. Scale bars are 50 µm in low-magnification images ('Merge' color panels) and 20 µm in high-magnification images (black and white panels). Bar graphs are presented as 'mean' plus 'standard deviation (SD)' here and in following figures unless specified. For the details of statistical methods please refer to Materials and methods and *Figure 2—source data 1*. Note that the confocal laser power used to image GFP (green in **B** and red in **C**) was 10x stronger for DTI-expressing brains (lower panels in **B** and **C**) compared to control brains (upper panels) here, and in all following ablation experiments.

The following source data and figure supplement are available for figure 2:

**Source data 1.** Statistical analysis of EB neuron ablation quantification.
**Figure supplement 1.** Ablation of pb-eb-gall and R2/R4m neurons lead to different defects.

Second, to ask whether R neurons are reciprocally required for pb-eb-gall small-field neuron dendrite elaboration in the EB we ablated R2/R4m neurons, which develop early in EB development, at early pupal stages. We observed a significant loss of R2/R4m GFP-labeled neurons following DTI expression (33.2% decrease at 24 hr ATS; 65.2% decrease at 48 hr ATS) (*Figure 2C*, *Figure 2—figure supplement 1C and F*, compare red arrows and arrowheads; quantification in *Figure 2D*). Upon the loss of a substantial fraction of R2/R4m neurons, the EB was reduced in size and pb-eb-gall dendrite areas were 29% smaller than in controls 48 hr ATS (*Figure 2—figure supplement 1C*, green circles; quantification in *Figure 2—figure supplement 1D*). In contrast, the areas occupied by pb-eb-gall axon terminals in the gall remained unchanged in control and in R2/R4m ablation animals (*Figure 2—figure supplement 1C*, blue arrow and arrowhead; quantification in *Figure 2—figure supplement 1E*). These observations show that R2/R4m axons are required locally for pb-eb-gall dendrite growth in the EB.

Third, to examine whether or not the ablation of outer R2/R4m axons affects the innervation pattern of inner R3 neuron axons in the EB, we labeled a subset of R3 neurons using *R54B05-lexA* while simultaneously ablating R2/R4m neurons (*Figure 2C* and *Figure 2—figure supplement 1F*). When R2/R4m neurons were ablated, R3 axon arbors occupied a larger area (26.6% increase) at 48 hr ATS; *Figure 2C* and *Figure 2—figure supplement 1F*, green ovals; quantification in *Figure 2E*). Therefore, R2/R4m ring neurons influence the size of R3 ring neuron axon arborizations.

Finally, to test whether other R neurons are similarly required for R2/R4m axon development, we used *R15F02-GAL4* to express DTI in R1/R3/R4d neurons. In contrast to what we observed following ablation of R2/R4m neurons, loss of >60% of GFP-labeled R1/R3/R4d neurons (with the remaining R1/R3/R4d neurons exhibiting >10 fold reduction in GFP expression) had no observable effects on EB morphology or R2/R4m axon expansion (data not shown). These data show that although R2/R4m axons constrain R3 axon expansion, other R neuron types likely do not serve a reciprocal function with respect to R2/R4m neurons.

These neuron ablation experiments suggest that pb-eb-gall dendrites and axons from different R neuron types play distinct roles in EB neuronal process development. The pb-eb-gall neurons are critical for the EB morphogenesis and axon patterning of all R types we examined. In addition, outer R2/R4m axons constrain inner R3 axon arbor expansion. Taken together, analysis of EB development and also EB cell-type-specific ablation experiments show that a series of spatiotemporally regulated interactions among EB neuronal processes facilitates the formation of proper R neuron axon laminae during the development of this central complex structure.

## Sema-1a and PlexA are expressed in the ellipsoid body

To understand how different groups of R neurons and pb-eb-gall neurons regulate the development of EB morphology and lamination, we investigated neuronal guidance molecules that could be involved in these processes. Using antibodies against the transmembrane semaphorin Sema-1a (*Yu et al., 1998*) and its receptor plexin A (PlexA) (*Sweeney et al., 2007*), we detected expression of both proteins in the central complex during early pupal stages (*Figure 3A and B*; *Figure 3—figure supplement 1A and B*). Both proteins are detected in the EB during pupal stages (*Figure 3A and B*, arrowheads), and Sema-1a is also found in the bulb (*Figure 3A*, arrows), where R neuron

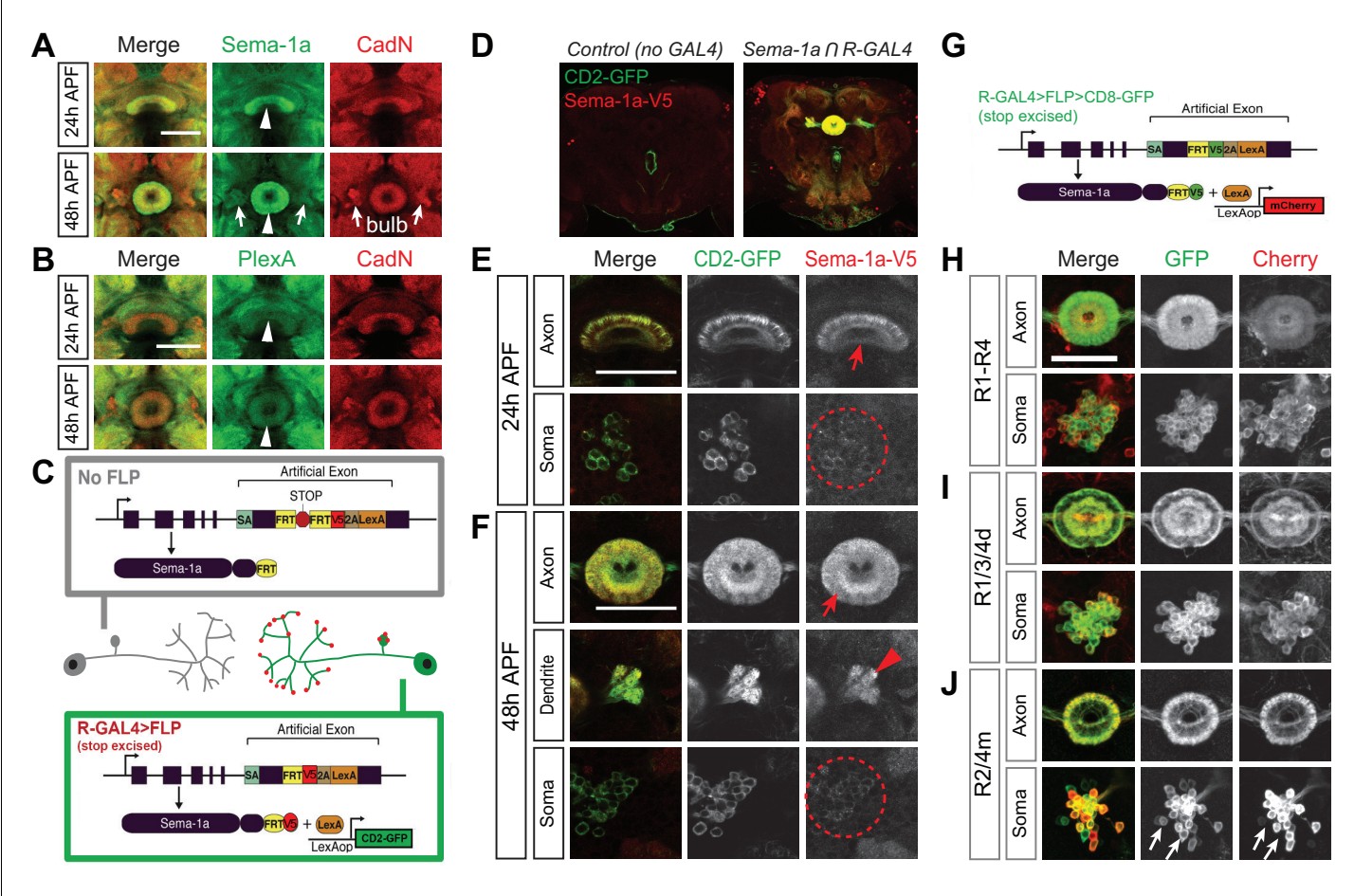

**Figure 3.** Ellipsoid body ring neurons express Sema-1a. (A and B) Antibody staining reveals that Sema-1a and PlexA are both expressed in the EB during early pupal stages (24 hr and 48 hr APF). Sema-1a is also detected in the bulb (arrows) at 48 hr APF. (C) Schematic (modified from: *Pecot et al., 2013*) shows our strategy for using cell-type-specific flipase (FLP) expression to conditionally tag endogenous Sema-1a with V5 epitopes, revealing Sema-1a expression while at the same time conditionally labeling these same neurons with CD2-GFP. (D–F) Both CD2-GFP and V5-tagged Sema-1a were conditionally expressed in R neurons using the strategy outlined in panel C when R neuron-specific *R11F03-GAL4* was used to express FLP. Low-magnification images show that CD2-GFP and V5 were not detected in animals when *R11F03-GAL4* was not present (D). High-magnification images reveal that V5-tagged Sema-1a is specifically enriched in R neuron axonal terminals within the EB at 24 hr APF (E) and at 48 hr APF, unlike CD2-GFP, which is uniformly localized in both R neuron soma and axons (F). Sema-1a-V5 exhibits strong expression throughout the EB, and it is also found in R neuron dendritic terminals in the bulb at 48 hr APF. (G) Schematic (modified from: *Pecot et al., 2013*) showing how Sema-1a-expressing R neurons are labeled by lexA-driving mCherry when FLP is expressed in these same R neurons using GAL4 drivers. (H–J) Using the strategy in panel G, multiple GAL4 drivers were used to label different groups of R neurons with CD8-GFP. Sema-1a-expressing R neurons, identified by mCherry expression, are shown at 48 hr APF. Scale bars are 50 μm.

The following figure supplement is available for figure 3:

**Figure supplement 1.** Sema-1a and PlexA are expressed in the central complex.

dendrites project. This suggests that EB neurons, particularly R neurons, express Sema-1a and PlexA during EB formation.

To examine Sema-1a expression in R neurons, we used a modified endogenous *Sema-1a* allele (*Sema-1a^FSF*) which includes a *FLP-stop-FLP* cassette inserted at the 3' end of *Sema-1a* coding sequence to conditionally tag Sema-1a and label Sema-1a-expressing neurons (*Pecot et al., 2013*). When the flipase (FLP) recombinase is expressed, the transcription termination ('stop') sequence is excised. This allows the V5 epitope to be added to the C-terminus of the endogenous Sema-1a protein and also results in co-transcription of *lexA* along with *Sema-1a* (*Figure 3C*). When FLP is expressed in all R neuron types using *R11F03-GAL4*, endogenous V5-tagged Sema-1a is detected at high levels in R axon terminals in the EB (*Figure 3E and F*, red arrows), but at low levels in the R neuron somas (*Figure 3E and F*, red circles). Consistent with our antibody staining, Sema-1a-V5 is also detected at R neuron dendrite terminals in the bulb (*Figure 3F*, red arrowhead). The broad distribution of Sema-1a antibody staining and Sema-1a–V5 in the EB at 48 hr APF suggests that Sema-1a expression is not restricted to specific R neuron types. To further assess Sema-1a expression in different R neuron types, *Sema-1a^FSF* was used to visualize Sema-1a expression in neurons following R neuron-specific FLP expression (*Figure 3G*). Indeed, dual labeling experiments using different GAL4 drivers (*R11F03-GAL4* for R1-R4; *R15F02-GAL4* for R1/R3/R4d; *EB-GAL4* for R2/R4m) confirmed that Sema-1a is expressed in most R neurons (*Figure 3I and J*, red) with the exception of a small subset of R2/R4m cells that are labeled only by GFP but not mCherry (*Figure 3J*, arrows).

These results show that Sema-1a is specifically recruited to the axons and dendrites of multiple R neuron types during the EB development, leading us to ask whether or not Sema-1a and PlexA influence R axon lamination in the EB.

## Sema-1a and PlexA are required for establishing ring neuron axon laminae in the ellipsoid body

Since *Sema-1a* is highly expressed in R neurons and *PlexA* is also expressed in the EB regions where R axons project, we examined loss-of-function (LOF) phenotypes in *Sema-1a* and *PlexA* null mutants (*Yu et al., 1998*; *Jeong et al., 2012*). Both *Sema-1a* and *PlexA* null mutants die during pupal stages, and in homozygous *Sema-1a* or *PlexA* mutant pupae the central complex, including the EB, displays severely disrupted morphologies (data not shown). Therefore, to investigate the functions of *Sema-1a* and *PlexA* in R neurons, we used GAL4/UAS-based RNA interference (RNAi) to knock down the expression of either gene in R neurons. For each gene at least two independent *UAS-RNAi* lines were used to control for possible off-target effects.

We examined the requirement for *Sema-1a* or *PlexA* in R2/R4m neurons. *R32H08-GAL4* was used to express GFP-tagged synaptotagmin (Syt-GFP) in order to reveal the circumferential distribution of R2/R4m presynaptic regions in the EB (*Figure 4Ai*, green). When *Sema-1a-RNAi* or *PlexA-RNAi* (*Pecot et al., 2013*; *Sweeney et al., 2007*) was expressed using *R32H08-GAL4*, the R2/4m ring (lamina) was partially disrupted, displaying a more scattered distribution of Syt-GFP labeling in the outer EB (*Figure 4Aii and 4Aiii*, green). By measuring the areas covered by Syt-GFP immunostaining, we found that the areas including R2/R4m synaptic labeling were increased by 32.1% and 27.1% in two different *Sema-1a-RNAi* lines, and by 19.3% and 22.8% in two different *PlexA-RNAi* lines, compared to controls (*Figure 4B*). Although the increase in area covered by *R32H08-GAL4>Syt-GFP* was less in *PlexA-RNAi* compared to *Sema-1a-RNAi* animals, the R2/R4m ring was more severely disrupted in *PlexA-RNAi*-expressing brains since we observed GFP-negative gaps and holes within the ring (*Figure 4Aiii*, arrowheads). These data reveal that both Sema-1a and PlexA are required for constraining R2/4m axon arbor growth in what appears to be an EB layer-autonomous manner.

## Full length Sema-1a is cell-autonomously required for constraining R4m axon arbors

To directly address whether *Sema-1a* is cell-autonomously required for regulating R2 and/or R4m axonal arbor elaboration, we generated *hsFLP* mosaic analysis with a repressible cell marker (MARCM) clones. First, we generated MARCM clones containing multiple (> or = to 5) R2 and/or R4m cells. Despite the presence of intermingled R2 and R4m axons in the EB, the composition of multiple-cell clones can be determined by the different locations of R2 and R4m dendrites in the bulb (*Figure 4—figure supplement 1A and B*, white and yellow circles). In control clones, R2/R4m

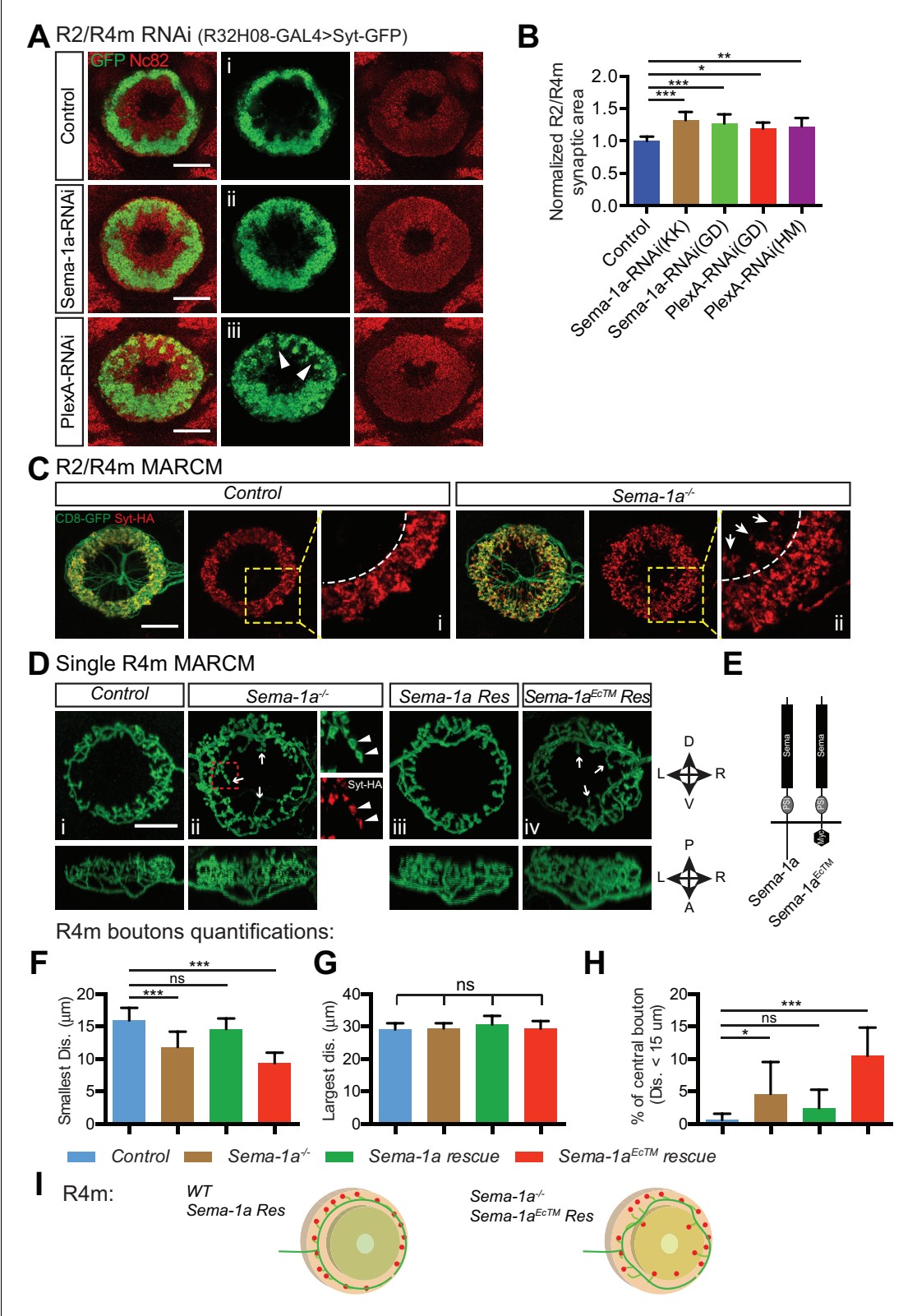

**Figure 4.** Sema-1a/PlexA signaling controls R2/R4m axon lamination and synapse localization in the ellipsoid body. (**A** and **B**) Pre-synaptic R2/R4m axon compartments were specifically labeled using *R32H08-GAL4*-driving GFP-tagged Synaptotagmin (Syt-GFP) in adult fly brains. Knocking down Sema-1a or PlexA in R2/R4m neurons led to aberrant ring neuron axon morphology and inward expansion of the R2/R4m synaptic domain in the EB. Syt-GFP areas were measured on Z-projection images and normalized to the mean area of the control group. Normalized values are compared in panel **B** for

*Figure 4 continued on next page*

*Figure 4 continued*

controls (1.000 ± 0.069 [n = 6 brains]), the two Sema-1a-RNAi groups (1.321 ± 0.129 [p<0.001] and 1.271 ± 0.140 [p<0.001] [n = 10 brains for each group]), and for the two different PlexA-RNAi groups (1.193 ± 0.091 [p=0.022, n = 7 brains] and 1.228 ± 0.127 [p=0.002, n = 12 brains]). Resource data for statistical analyses are available in *Figure 4—source data 1*. (C) MARCM clones were generated using hsFLP. R2/R4m GAL4 drivers were used to label R2/R4m neurons with CD8-GFP in adult flies. HA-tagged synatotagmin (Syt-HA) was co-expressed to label pre-synaptic specializations in *WT* and *Sema-1a^{-/-}* clones. In a *Sema-1a^{-/-}* clone containing multiple R2 and R4m cells, Syt-HA-labeled axon terminals expanded inwardly within the EB, recapitulating the *Sema-1a-RNAi* phenotype in panel A. (D–E) Frontal- (upper panels) and dorsal- (lower panels) view images of axon arbors of single R4m MARCM clones reveal that R4m axons become more complex and some R4m boutons are located closer to the EB center in *Sema-1a^{-/-}* and *Sema-1a^{EcTM}* rescue clones than in control and *Sema-1a rescue* clones. Schematics to the right show the brain coordinates and domain organization of WT Sema-1a and Sema-1a^{EcTM} proteins (E). For rescue experiments, full length Sema-1a or truncated Sema-1a lacking its cytoplasmic domain (Sema-1a^{EcTM}) were expressed in *Sema-1a^{-/-}* clones consisting of R4m neurons. (F–H) The pre-synaptic boutons of R4m neurons in the EB were manually determined based on their enlarged morphologies as observed in the reconstructed 3D images of GFP-labeled R4m axon arbor (see *Figure 4—figure supplement 1C*). The distances between each bouton and the center of EB canal were measured. The smallest and largest distances were plotted and compared to controls in panels F and G, respectively. The percentage of R4m boutons that were no more than 15 µm away from the EB center is shown in panel H. n = 12 Single Cell Clones (SSCs) for controls; n = 16 SSCs for *Sema-1a^{-/-}* group; n = 9 SSCs for *Sema-1a rescue*; n = 10 SSCs for *Sema-1a^{EcTM}* rescue. Detailed statistical analyses are available in *Figure 4—source data 2*. (I) Schematics highlight R4m axon arbor changes shown in panel D. 'ns' p>0.1234; *p<0.0332; **p<0.0021; ***p<0.0002. Scale bars are 20 µm.

The following source data and figure supplements are available for figure 4:

**Source data 1.** Statistical analysis of R2/R4m syt-GFP quantification.

**Source data 2.** Statistical analysis of single R4m MARCM.

**Figure supplement 1.** *Sema-1a* is required cell-autonomously for R4m, but not R2, axon arbor and synapse development.

**Figure supplement 2.** PlexA and Sema-1a are differentially required in R1–3 neurons for EB morphogenesis and R axon lamination.

synaptic terminals, labeled by HA-tagged synaptotagmin (Syt-HA), form a regular ring within the peripheral EB (*Figure 4C*, *Control*), displaying a clear inner boundary (*Figure 4Ci*, dashed line). However, in about half of *Sema-1a* mutant clones, the R2/R4m ring was partially deformed (*Figure 4C*, *Sema-1a^{-/-}*). A significant fraction of Syt-HA-positive boutons were mis-localized to regions closer to the EB central canal (*Figure 4Cii*, arrows) and beyond the normal R2/R4m inner boundary (*Figure 4Cii*, dashed lines). This is in line with the lamination defects observed when *Sema-1a* is knocked down in these same R neuron types (*Figure 4A*). Interestingly, this outer EB lamination phenotype was only observed in Sema-1a^{-/-} clones consisting of R4m neurons (*Figure 4—figure supplement 1A*), but not in clones composed of only R2 cells (*Figure 4—figure supplement 1B*). These results show that R2/R4m axon lamination defects in *Sema-1a* LOF experiments are largely due to lack of Sema-1a in R4m neurons, underscoring the select interactions among ring neurons that are required to generate EB laminar organization.

To confirm the cell-autonomous function of *Sema-1a*, we generated smaller clones in order to label single R4m neurons (*Figure 4D*). Indeed, compared to control R4m neurons (*Figure 4Di*), single *Sema-1a^{-/-}* R4m neurons had aberrant axon arbor morphology (*Figure 4Dii*). Further, some synaptic boutons, identified by both GFP-labeled varicosities and Syt-HA labeling (*Figure 4Dii*, small panels, arrowheads), were localized to more central EB regions (*Figure 4Dii*, arrows). To confirm that these R4m phenotypes resulted from loss of *Sema-1a* function, we expressed a full-length *Sema-1a* (*Sema-1a rescue*) or a truncated *Sema-1a* (*Sema-1a^{EcTM} rescue*) transgene in *Sema-1a^{-/-}* R4m cells (*Figure 4E*). Expression of *Sema-1a*, but not *Sema-1a^{EcTM}*, in *Sema-1a^{-/-}* R4m cells rescued R4m axon arbor phenotypes (*Figure 4Diii and 4Div*). To quantify changes in R4m synapse localization, we counted R4m pre-synaptic boutons based on the presence of CD8-GFP^{+} varicosities in reconstructed 3D images (*Figure 4—figure supplement 1Ciii and 1Cvii*, white dots). We then measured the distance from each bouton to the EB canal center (*Figure 4—figure supplement 1Civ and 1Cviii*, color-coded bouton distance), grouped the smallest and largest distances between R4m boutons and the EB center in single R4m clones, and compared them across all four genotypes. The smallest distances, which indicate how close R4m boutons are to the EB center, were significantly decreased in *Sema-1a^{-/-}* (11.85 ± 2.36 µm, n = 16 singe-cell clones [SSCs]) and in *Sema-1a^{EcTM} rescue*

(9.34 ± 1.64 μm, n = 10 SSCs) neurons but showed no difference compared to control R neurons (15.95 ± 1.93 μm, n = 12 SSCs) or *Sema-1a rescue* R neurons (14.62 ± 1.64 μm, n = 9 SSCs) (*Figure 4F*). In contrast, the largest distances from the EB center, which are a measure of how wide the R4m ring is, were the same across all four genotypes (*Figure 4G*).

R2/R4m synapses are normally excluded from the central EB region, which is innervated by R3 axons that have arbors with a minimum diameter of about 30 μm (*Figure 5—figure supplement 1G*, 'R3 Diameter'). We scored central R4m boutons (defined by their locations within a 15 μm radius originating at the EB center) and calculated the percentage ratio of central bouton number *vs* total bouton number for each R4m neuron (*Figure 4H*). A significantly larger fraction of R4m boutons was found in the central EB region in both *Sema-1a*[-/-] (4.66%, n = 16 SSCs) and *Sema-1a*[EcTM] *rescue* (10.52%, n = 10 SSCs) compared to the control group (0.62%, n = 12 SSCs). The *Sema-1a rescue* R4m neurons (2.43%, n = 9 SSCs), although they exhibited 32.3% fewer total synapses (*Figure 4—figure supplement 1D*), showed no significant change in central synapse distribution compared to controls. In addition to these synapse distribution changes, R4m axon arbor morphology changes were also quantified by tracing R4m axon branches (*Figure 4—figure supplement 1Cii and 1Cvi*). Total branch length was increased in *Sema-1a*[-/-] (11.6% increase) and *Sema-1a*[EcTM] *rescue* (16.5% increase) (*Figure 4—figure supplement 1E*), and the total number of branch points was decreased by 26.9% in *Sema-1a rescue* (*Figure 4—figure supplement 1F*), compared to controls, suggesting that Sema-1a regulates R4m axon branch formation and extension.

Taken together, these results demonstrate that full length Sema-1a, but not truncated Sema-1a lacking its cytoplasmic domain, is required cell-autonomously to regulate R4m axon branch growth, lamination and synapse distribution within the EB. Therefore, these functions are likely mediated through a Sema-1a 'reverse signaling' pathway in which the Sema-1a protein functions as a receptor in R4m neurons and PlexA protein servers as its ligand in other EB neurons (*Battistini and Tamagnone, 2016*).

Since lamination defects were observed when either Sema-1a or PlexA was knocked down in R2/R4m (*Figure 4A*), we suspected that R2 neurons express PlexA and therefore regulate R4m axon lamination through Sema-1a reverse signaling. Unfortunately, all the available R2 drivers we characterized also drive expression in R4m neurons. However, *R40G10-GAL4* drives expression in R1, R2 and R3 neurons throughout pupal and adult stages (*Lovick et al., 2017*). Therefore, we used this driver to knock down Sema-1a or PlexA in R1–3 neurons, and we also used *R34D03-lexA* to label a subset of R4m axons.

Down-regulation of Sema-1a or PlexA using *R40G10-GAL4* resulted in different EB morphology and R axon elaboration phenotypes. In *Sema-1a-RNAi*–expressing animals, the EB partially lost its circular shape. R4m axons formed an irregular ring but still appeared organized in a laminar fashion in EB outer regions (*Figure 4—figure supplement 2, Aii and Av*). More severe phenotypes were observed in *PlexA-RNAi* expressing animals. In many *R40G10>PlexA-RNAi* animals, the EB remained completely open and R4m axons expanded their axon projections to cover most EB regions (*Figure 4—figure supplement 2, Aiii and Avi*). Taken together, our data suggest that PlexA in R1–3, and most likely in R2 alone since knocking down PlexA in R1/R3 with multiple drivers did not generate similar defects (data not shown), plays an important role in regulating R4m axon lamination formation through Sema-1a reverse signaling pathway.

## Sema-1a and PlexA in R2/R4m neurons are required non-cell autonomously for R3 axon expansion

From our EB neuron ablation experiments during pupal development, we learned that R2/R4m axons constrain R3 axon expansion within the EB. To further investigate how the disruption of R2/R4m axon lamination in *Sema-1a* or *PlexA* LOF mutants affects R3 axons, we used the *R54B05-lexA* driver to express mtdT and label a subset of R3 neurons, while we also used the *R32H08-GAL4* driver to express CD8-GFP and *Sema-1a-RNAi* (or *PlexA-RNAi*) in R2/R4m neurons (*Figure 5A*). In controls, R2/R4m and R3 axon arbors exhibit characteristic sizes and are adjacent to one another with a clear boundary separating these two R neuron arbors (*Figure 5Ai and 5Aiv*, yellow dashed line). However, in either R2/R4m *Sema-1a-RNAi* or *PlexA-RNAi*–expressing fly lines, R3 axon arbors expand outward within the EB (*Figure 5Av and 5Avi*), and this is accompanied by inward expansion of R2/R4m axons (*Figure 5Aii and 5Aiii*). This results in disruption of the boundary between these two laminae, with the outer edge of the R3 axon arbors displaying a jagged morphology and R3

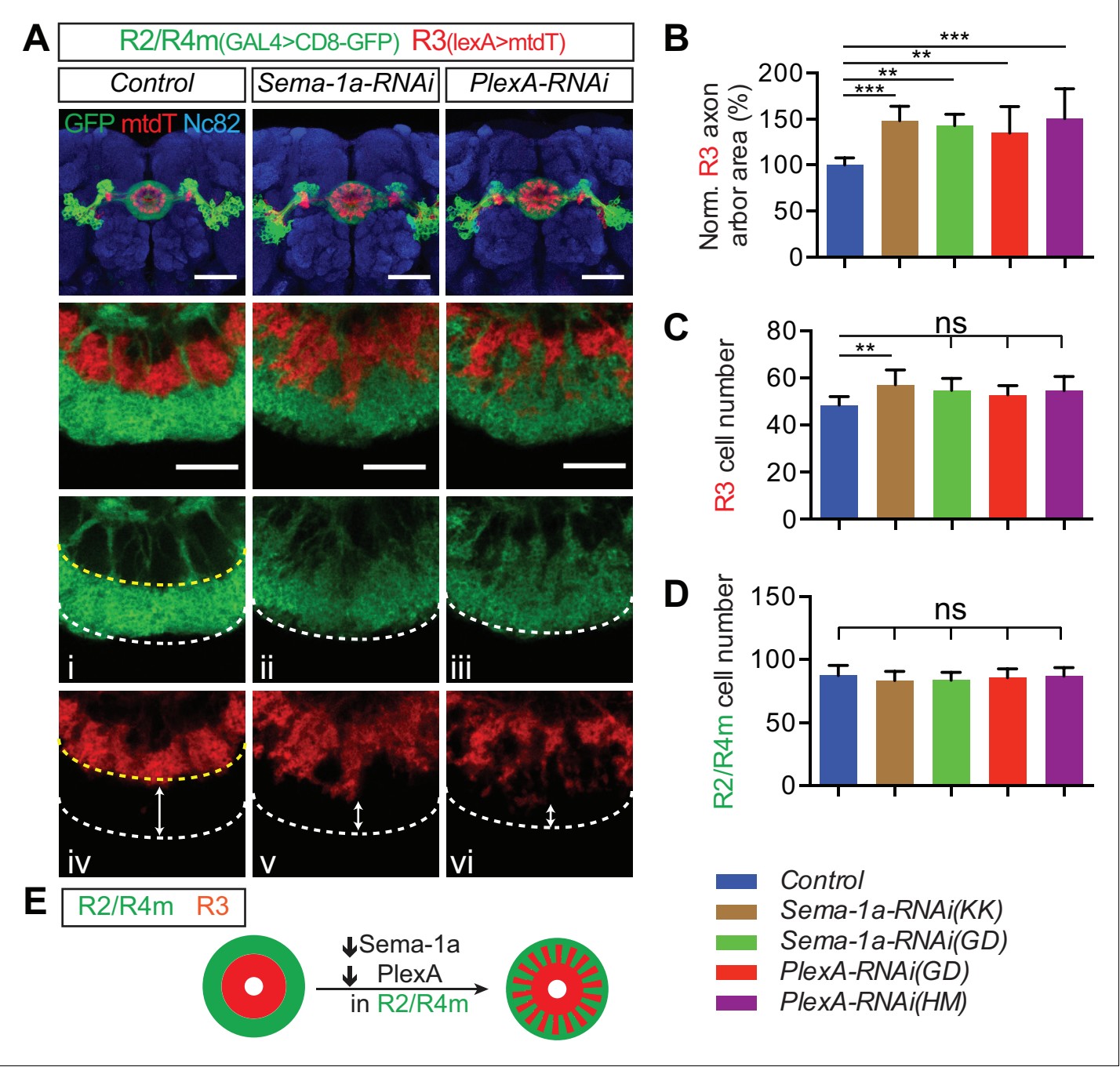

**Figure 5.** Sema-1a and PlexA are required in R2/R4m neurons to constrain R axon growth within the ellipsoid body. (A–D) R2/R4m neurons were labeled by *R32H08-GAL4*-driving CD8-GFP, while R3 axons were labeled using *R54B05-lexA*-driving mtdTomato (mtdT) in adult fly brains. Knocking down Sema-1a or PlexA in R2/R4m neurons did not change axon trajectory or cell numbers of either R3 or R2/R4m (panels **C** and **D**), but resulted in non-cell autonomous R3 axon arbor expansion in the EB (panel **B**). Seven to eight brains for each genotype were used for quantification. 'ns' p>0.1234; *p<0.0332; **p<0.0021; ***p<0.0002. See *Figure 5—source data 1* for detailed statistical analyses. (**E**) Schematics showing that both R3 (red) and R2/R4m (green) axons expand when Sema-1a or PlexA is down-regulated in R2/R4m, leading to intermingled R3 and R2/R4m axons. Scale bars are 50 μm in low-magnification images in top panels and 20 μm in high-magnification images in other panels.

The following source data and figure supplements are available for figure 5:

**Source data 1.** Statistical analysis of R3 axon quantification in R2/4m RNAi.

**Source data 2.** Statistical analysis of R3 RNAi quantification.

*Figure 5 continued on next page*

*Figure 5 continued*

**Source data 3.** Statistical analysis of single R3 MARCM quantification.

**Figure supplement 1.** R2/R4m Sema-1a, but not R3 Sema-1a/PlexA, is required for R axon lamination.

**Figure supplement 2.** Sema-1a and PlexA are not required for pb-eb-gall dendrite elaboration and synaptogenesis in the EB.

axons extending outward to more peripheral EB regions. Further, the smallest distance between R3 axons and the outer edge of the R2/R4m ring was greatly decreased (*Figure 5Aiv, 5Av and 5Avi*; double-headed arrows). These changes in R axon patterning appear to develop gradually during EB formation (*Figure 5—figure supplement 1A and B*).

We quantified the expansion of R3 axon arbors by measuring the area covered by mtdT immunostaining within the EB in adult brain Z-projection images. Following normalization to controls (n = 8 animals), the area occupied by R3 axon arbors increased 48.3 ± 15.6% (n = 8 brains) and 42.8 ± 12.4% (n = 7 brains) for the two independent *Sema-1a-RNAi* lines, and 34.8 ± 28.7% (n = 8 brains) and 50.4 ± 32.6% (n = 8 brains) for the two independent *PlexA-RNAi* lines (*Figure 5B*). To address whether Sema-1a or PlexA are required for controlling R neuron numbers, we quantified R3 and R2/R4m cell numbers using mtdT and CD8-GFP labeling, respectively. R3 numbers were similar to controls in all of the RNAi lines except for a small increase (17.6%) in one *Sema-1a-RNAi* line (*Figure 5C*). Further, we found that R2/R4m cell numbers remained unchanged in all RNAi experiments compared to the controls (*Figure 5D*). To further examine the role of R2 and R4m in constraining R3 axon growth, *R40G10-GAL4* was used to express *Sema-1a* or *PlexA-RNAi* in R1–3 neurons. R54B05-lexA>mtdT-labeled R3 axon arbors were mildly altered but remained within their territory in the inner EB in *R40G10>Sema-1a-RNAi* animals (*Figure 4—figure supplement 2, Bii*). However, R3 axons generally increased their elaboration within the EB in *R40G10>PlexA-RNAi* animals (*Figure 4—figure supplement 2, Biii*). Taken together, these results support the idea that Sema-1a/PlexA-dependent R2/R4m axon patterning is required to control R3 axon expansion through a local repellent signal, thereby regulating R axon lamination.

Both Sema-1a and PlexA can signal as receptors to mediate repulsion (*Jongbloets and Pasterkamp, 2014*; *Battistini and Tamagnone, 2016*). We wondered whether Sema-1a or PlexA, given their expression patterns, act as signals or receptors between R2/R4m and R3 axons. Therefore, we next tested whether Sema-1a or PlexA function cell-autonomously in R3 neurons to control R3 axon expansion. First, we found that knocking down *Sema-1a* or *PlexA* in R3 neurons did not change the size or morphology of the R3 rings (*Figure 5—figure supplement 1C*; quantification in *Figure 5—figure supplement 1D*). Second, MARCM experiments show that single R3 axon ring diameters are not different between control and *Sema-1a$^{-/-}$* R3 neurons (*Figure 5—figure supplement 1F*; 'Diameter' quantification in *Figure 5—figure supplement 1G*). However, in *Sema-1a$^{-/-}$* single R3 MARCM clones, many R3 neurons elaborate axon branches that grow more posteriorly within the EB, and for some R3 neurons (6 out of 21 cells) out of the EB and into the posterior FB (*Figure 5—figure supplement 1E and F*; quantification of 'R3 Thickness' in *Figure 5—figure supplement 1G*). This suggests that Sema-1a controls longer range R axon extension to secure the overall separation of neighboring brain structures. We were unable to use MARCM to test the cell autonomy of PlexA in R neurons since PlexA is located on the fourth chromosome and tools are not presently available for this experiment.

Together, these results show that it is unlikely PlexA or Sema-1a act in R3 as receptors for R2/R4m Sema-1a or PlexA to directly regulate R3 axon growth. Although we find that Sema-1a and PlexA function in R2/R4m neurons to organize axon lamination, our data suggest that R2/R4m axons use other molecules to constrain R3 axons and refine axon patterning and lamination during pupal development.

## Sema-1a and PlexA are not required for pb-eb-gall Dendrite development in the EB

Our analyses of central complex development show that pb-eb-gall dendrites play an important role in controlling R axon lamination (*Figure 1—figure supplement 1K* and *Figure 2B*). Is R axon lamination also required for pb-eb-gall dendrite elaboration, and do Sema-1a or PlexA play a role in pb-eb-gall development? To address these issues, we first used *R19G02-lexA* to drive mtdT expression in pb-eb-gall neurons in combination with *R32H08-GAL4* to knock down Sema-1a or PlexA in these neurons. The pb-eb-gall dendrites appeared normal when R2/R4m axons expanded their elaboration within the EB in either *Sema-1a-RNAi* or *PlexA-RNAi*–expressing animals (*Figure 5—figure supplement 2, Aiv-vi*). Secondly, we used the syb:GRASP technique (*Macpherson et al., 2015*) to detect potential synaptic connections between R2/R4m axons and pb-eb-gall dendrites. The spGFP$_{1-10}$-tagged neuronal synaptobrevin (syb-spGFP$_{1-10}$) was expressed in R2/R4m by *R32H08-GAL4* while CD4-spGFP$_{11}$ and mtdT were expressed in pb-eb-gall neurons. Reconstituted GFP (rcGFP) was detected in the EB in both control and *UAS-Sema-1a-RNAi*-expressing animals, even though different GFP fluorescence patterns were observed (*Figure 5—figure supplement 2B*). These data suggest that R axon lamination defects do not affect pb-eb-gall dendrite elaboration or synaptogenesis between R axons and pb-eb-gall dendrites. Finally, *R19G02-GAL4* was used to drive *Sema-1a-RNAi* or *PlexA-RNAi* in pb-eb-gall neurons. Knocking down Sema-1a or PlexA in pb-eb-gall neurons partially or completely re-directed their axon projections from the gall (*Figure 5—figure supplement 2, Ci-iii and Di-iii*, arrows) to a region close to the EB (*Figure 5—figure supplement 2, Ci-iii and Di-iii*, arrowheads). However, loss of Sema-1a or PlexA in pb-eb-gall neurons did not affect pb-eb-gall dendrite elaboration or R2/4 m and R3/R4d axon lamination (*Figure 5—figure supplement 2, Civ-vi and Div-vi*), suggesting that Sema-1a and PlexA in pb-eb-gall neurons are not required for pb-eb-gall dendrite or R axon development in the EB.

## Ring neurons recruit GABA and GABA-A receptors to their axon terminals

In invertebrate and vertebrate visual systems, pre- and post-synaptic neurites co-stratify to facilitate appropriate connections for visual system function (*Zipursky and Sanes, 2010*; *Zhang et al., 2017*). For example, co-stratification of axons from select On cone bipolar cells with dendrites of On-Off direction selective ganglion cells in specific sub-laminae of the inner plexiform layer (IPL) is critical for functional direction-selective responses in mice (*Duan et al., 2014*). In the EB, multiple different types of laminated R axons converge onto pb-eb-gall neuron dendrites, and so this raises the question: what function does R axon lamination serve, and might it play a role in specifying distinct synaptic connections among R neurons? To understand the functional significance of Sema-1a/PlexA-mediated R neuron axon lamination, we first investigated the neurochemical properties of R neurons.

Abundant protein expression of gamma aminobutyric acid (GABA) and RDL (a GABA-A receptor subunit) has been observed in the EB and appears to be closely associated with R neuron axons (*Kahsai et al., 2012*; *Martín-Peña et al., 2014*; *Enell et al., 2007*). However, the sources of GABA and RDL and their functions in the EB are largely unknown, although a group of R2/R4m neurons are known to be GABAergic and thermogenetic activation of these neurons impairs medium-term olfactory memory (*Zhang et al., 2013*). To address this issue, we took advantage of newly developed protein-trap GAL4 and QF2 drivers (*Diao et al., 2015*). These drivers were obtained by converting a MiMic insertion in a coding intron to an exogenous exon containing coding sequences for the transcriptional activators GAL4 or QF2 (*Venken et al., 2011*). This strategy reliably represents the expression of target genes, often more faithfully than traditional enhancer-trap GAL4 driver lines (*Diao et al., 2015*).

We found that fluorescent reporters driven by *Gad1$^{MI}$-QF2* and *Rdl$^{MI}$-GAL4* (*Diao et al., 2015*) were both expressed in a large number of R neurons, displaying characteristic axon trajectories into the bulb and EB (*Figure 6A and D*, arrowheads and arrows). Expression of either *UAS-Gad1-RNAi* or *UAS-Rdl-RNAi* in R2/R4m neurons using *R32H08-GAL4* decreased GABA and RDL immunoreactivity, respectively, in the peripheral EB, the location where R2/R4m axons project (*Figure 6B* arrowhead; *Figure 6E*, arrows). Loss of GABA or RDL following RNAi knockdown was quantified by measuring the area of strong immunostaining revealed by each antibody in the EB (*Figure 6B and*

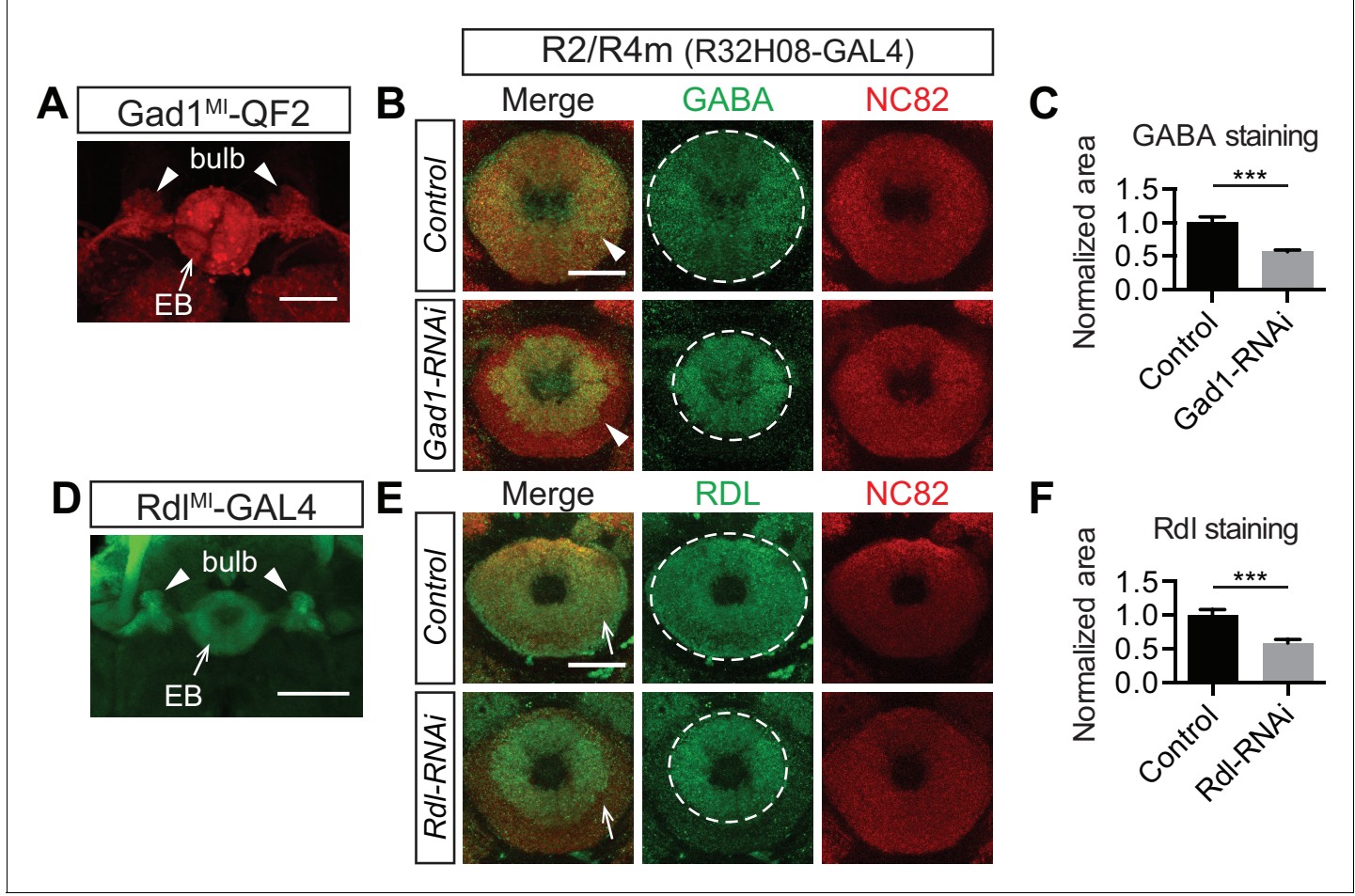

**Figure 6.** Ring neurons transport GABA and GABA-A receptors to their axonal terminals in the ellipsoid body. (A and D) MiMic-based GAL4 and QF2 drivers (*Diao et al., 2015*) were used to express fluorescent reporters and to access Gad1(A) and Rdl (D) expression in adult fruitfly brains. Both drivers clearly express in a large group of ring neurons, which project from the lateral regions into the bulb (arrowheads) and EB (arrows). (B, C, E, F) GABA and Rdl immunostaining is enriched in the EB in adult fruitfly brains. Knocking down Gad1 and Rdl in R2/R4m neurons using *R32H08-GAL4* significantly decreased GABA (arrowheads in panel B) and Rdl (arrows in panel E) immunostaining in the peripheral regions of the EB. The area covered by strong GABA and Rdl immunostaining in the EB was measured using maximal intensity Z-projection images. All measurements were normalized to the mean of the control groups, and they were compared between control and RNAi groups as shown in panels C and F. For GABA staining in the control group, normalized area was 1.000 ± 0.083 (n = 5), and for the RNAi group the normalized area was 0.563 ± 0.028 (n = 5) and significantly reduced, compared to the control (p=0.0079). For Rdl staining, the normalized area was 1.000 ± 0.087 (n = 5) and for the RNAi group the normalized area was 0.587 ± 0.052 (n = 5), again, significantly reduced compared to the control (p=0.0079). Detailed statistical analyses are available in *Figure 6—source data 1*. Scale bars are 50 µm in panels A and D, and 20 µm in panels B and E.

The following source data and figure supplement are available for figure 6:

**Source data 1.** Statistical analysis of Gad1 and Rdl RNAi in R2/R4m quantification.

**Figure supplement 1.** Ellipsoid body ring neurons are not cholinergic, glutamatergic or dopaminergic.

*E*, circles; quantification in *Figure 6C and F*). Expression levels of GABA and RDL were also reduced in central EB regions when R3 Gal4 drivers were used to express these corresponding RNAi transgenes (data not shown). We also examined three other common neurotransmitter pathways (acetylcholine, glutamate and dopamine) in the EB using antibody staining and similar genetic labeling strategies as we used for GABA. Many fewer cholinergic, glutamatergic or dopaminergic inputs were observed in the EB, and most of these were preferentially localized to posterior EB regions (*Figure 6—figure supplement 1*, arrows). Taken together, these results show that multiple types of

R neurons are GABAergic and also that they recruit GABA-A receptors to their axon terminals in the EB, raising the question of whether R axon lamination influences EB inhibitory synaptic properties.

## Sema-1a dependent R neuron axon lamination reduces axon-axon inhibition across adjacent ellipsoid body rings

The co-recruitment of GABA and GABA-A receptors to R neuron axon terminals suggests that local inhibitory circuits are formed among R axon terminals. R neuron axons are precisely organized into discrete rings/laminae, suggesting that inhibitory synaptic connections among R neuron axons may also be organized in a laminated fashion such that co-stratified R neuron synaptic terminals inhibit each other. This leads to the prediction that R neuron axons that project within adjacent rings, such as R2/R4m and R3, do not exhibit coupled inhibitory interactions.

To test this idea, we took advantage of the R axon lamination phenotype we observed in *Sema-1a* LOF mutants. We used the *R32H08-GAL4* driver to express *UAS-mCherry* in R2/R4m neurons in order to locate their R2/R4m cell bodies for electrophysiological recording, and also to express *Sema-1a-RNAi* in these same R neurons and disrupt R axon lamination. Additionally, the *R54B05-lexA* driver was used to express *lexAop2-CsChrimson-mVenus* in a subset of R3 neurons, allowing us to control optogenetic stimulation of R3 neurons (*Figure 7B*). Consistent with the anatomical defects we observed in R neurons in the absence of Sema-1a (*Figures 4* and *5*), expression of *Sema-1a-RNAi* in R2/R4m resulted in general lamination defects of both R2/R4m and R3 rings (*Figure 7A*, red and green) and also morphological changes in the axon arbors of single R2/R4m neurons that we could identify with dye-injections following electrophysiological recordings (*Figure 7Ai and 7Aii*, white).

To examine inhibitory currents among R3 and R2/R4m axons, we used optogenetic stimulation coupled with electrophysiological recording in an ex vivo fly brain preparation (*Figure 7B*) (*Inagaki et al., 2014*; *Liu et al., 2016*). CsChrimson depolarizes R3 neurons upon LED light stimulation, which we first confirmed by field recordings near R3 somas (data not shown). To detect inhibitory input onto R2/R4m neurons from R3 neurons, we performed current-clamp recordings on the cell bodies of single R2 or R4m neurons. In control animals (n = 4), injection of current above a certain threshold (10.5 ± 5.7 pA) successfully evoked trains of action potentials (APs) in R2/R4m neurons (*Figure 7C*, blue line). We plotted AP frequencies against the injected current amplitudes (*Figure 7D,F–I* curves) and observed that above the current threshold the AP frequency in R2/R4m neurons was positively correlated with the depolarization current amplitudes. More importantly, LED stimulation of R3 neurons had little effect on evoking APs in R2/R4m neurons in these control animals (*Figure 7C and D*, red vs blue lines). The slopes of the *f-I* curves were no different when the LED light was off or on in control brains (0.142 ± 0.032 Hz/pA for LED-off *vs* 0.125 ± 0.032 Hz/pA for LED-on, p=0.391) (*Figure 7E*), showing that the excitability of R2/R4m neurons is not coupled to R3 activation when R2/R4m and R3 axons are physically separated into two adjacent laminae in WT animals.

In *Sema-1a-RNAi* animals (n = 4), APs were also successfully evoked when depolarization currents above the threshold (9.8 ± 6.6 pA) were injected into the recorded R2/R4m neurons (*Figure 7F and G*, blue lines). However, following R3 neuron activation (LED on) in these *Sema-1a-RNAi* animals, R2/R4m neurons generated many fewer APs when evoked with the same depolarization currents above the current threshold (*Figure 7F and G*, red vs blue lines). In contrast to controls, the slopes of *f-I* curves were significantly decreased in *Sema-1a-RNAi* expressing animals when the LED light was on (0.208 ± 0.074 Hz/pA for LED-off *vs* 0.05 ± 0.019 Hz/pA for LED-on, p=0.023) (*Figure 7H*), showing that it is more difficult to evoke APs in R2/R4m neurons when light-activated R3 axons are intermingled with R2/R4m axons in *Sema-1a-RNAi* brains.

To further assess how optical R3 neuron activation inhibits R2/R4m neurons, we calculated the resting potential (Vm) and input resistance (Rin) at the R2/R4m membrane. Consistently, LED stimulation of R3 resulted in no change of Vm (−56.45 ± 5.16 mV for LED-off *vs* −57.14 ± 6.33 mV for LED-on, p=0.546) or Rin (1.105 ± 0.157 GOhm for LED-off *vs* 1.120 ± 0.148 GOhm for LED-on, p=0.822) in R2/R4m neurons in control animals (*Figure 7—figure supplement 1A and B*). However, in *Sema-1a-RNAi*-expressing animals, R3 activation significantly hyperpolarized the Vm (−58.60 ± 1.20 mV for LED-off *vs* −63.26 ± 2.69 mV for LED-on, p=0.031) and decreased the Rin of R2/R4m neurons (1.001 ± 0.299 GOhm for LED-off vs 0.803 ± 0.365 GOhm for LED-on, p=0.014) (*Figure 7—figure supplement 1C and D*). These results indicate that R3 activation acutely altered

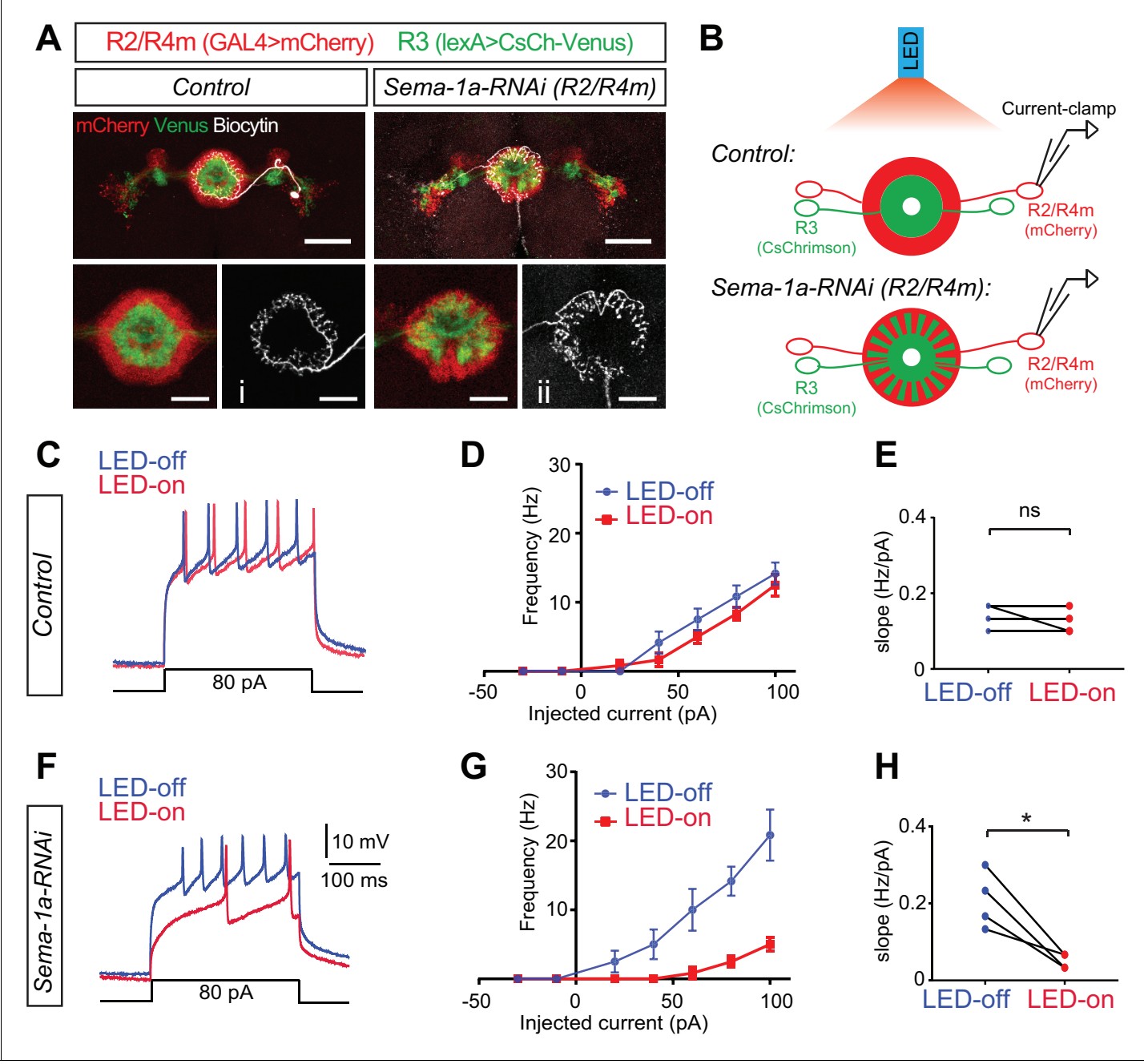

**Figure 7.** Loss of *Sema-1a* in R2/R4m neurons increases R2/R4m inhibition in response to R3 neuron activation. (**A**) In the photo-stimulation and recording experiments, R2/R4m neurons were labeled by *R32H08-GAL4*-driving mCherry (red) and R3 neurons expressed CsChrimson-Venus (green) driven by *R46D01-LexA* in adult fly brains. Note the separation of R3 and R2/R4m rings in the control brain (left) and intermingling of R3 and R2/R4m rings in the *R32H08-GAL4*-driving *Sema-1a-RNAi* brain (right). Injection of Biocytin after recording and Alexa647-Streptavidin staining revealed the morphology and identify of recorded R2 and R4m neurons. Scale bars are 50 μm in large panels and 20 μm in small panels. (**B**) Diagrams highlight the experimental design of photostimulation of R3 neurons and patch-clamp recording of single R2 or R4m neurons in ex vivo cultured adult fruitfly brains. Control and RNAi flies share all the transgenes for R3 neuron optogenetic manipulation and R2/R4m neuron labeling, except that *Sema-1a-RNAi* was expressed in R2/R4m neurons in RNAi flies. (**C**) Example traces show that current injections evoked neuronal spiking in R2/R4m neurons in a control fruitfly brain. Blue lines indicate voltage trace without LED light and red lines indicate voltage trace with LED light. (**D**) *F-I* curve of the R2/R4m cells from control R2/R4m cells with (red) or without (blue) LED light stimulation of R3 neurons (n = 4 cells from 4 animals). Data are shown as mean±SEM. (**E**) Comparisons of *F-I* slopes in D. Each symbol represents one cell (n = 4). LED-off: 0.142 ± 0.032 Hz/pA; LED-on: 0.125 ± 0.032 Hz/pA; p=0.391. (**F–H**) Similar to C-E for Sema-1a-RNAi expressing R2/R4m cells (n = 4). In panel H, LED-off: 0.208 ± 0.074 Hz/pA; LED-on: 0.050 ± 0.019 Hz/pA; p=0.0234. Raw recording data and statistical analyses are available in the Zip file for *Figure 7—source data 1*.

*Figure 7 continued*

The following source data and figure supplement are available for figure 7:

**Source data 1.** Raw data and statistical analysis of electrophysiological recording.

**Figure supplement 1.** Electrophysiological characterization of R3 and R2/R4m axon-axon inhibition.

the electrophysiological properties of R2/R4m cell membranes when Sema-1a-mediated axon lamination was disrupted.

Taken together, these data support the idea that when Sema-1a-mediated R axon lamination is disrupted, R3 and R2/R4m axons exhibit ectopic contacts, aberrantly coupling their activity through ectopic inhibitory synapses. Therefore, laminar organization of R axons plays an important role in specifying inhibitory synapse formation between different types of R axons in the EB.

## Discussion

The insect brain central complex (CX) is composed of densely laminated neuropil structures, including the FB and the EB. The *Drosophila* EB harbors a particular laminar organization in which axons from multiple ring (R) neuron types form several adjacent concentric rings/layers. We show here that pb-eb-gall neuron dendrites sequentially associate with ring neuron R2/R4m and R3/R4d axons within the EB during early pupal stages. The pb-eb-gall neurons play essential roles in directing R neuron axon patterning, and R2/R4m neurons are required for confining R3 axon expansion within correct EB laminae. The transmembrane semaphorin Sema-1a and also the PlexA protein each function cell-autonomously in R2/R4m neurons and non-cell autonomously in neighboring R3 neurons to prevent ectopic mixing of these normally separated R neuron axons. Importantly, Sema-1a-dependent R axon lamination is necessary to prevent aberrant inhibitory synapse formation between R2/R4m and R3 axons, as revealed by optogenetic stimulation of R3 neurons and electrophysiological recording from R2/R4m neurons. Our results define interactions among R neurons that are important for lamination within the EB during pupal development, and they demonstrate that EB lamination is critical for constraining inhibitory synapse formation to specific EB laminae.

### Interactions between small- and large-field neurons during EB laminae formation

Previous anatomical analyses show that small-field pb-eb-gall neurons and large-field R neurons elaborate their dendrites and axons, respectively, in the anterior region of the EB (*Wolff et al., 2015*). How small- and large-filed neuron process elaboration is coordinated during pupal development so as to establish laminar and columnar organization during EB formation was not known. Our findings suggest that small-field pb-eb-gall neuron dendrites and large-field R neuron axons locally interact during the EB lamination.

The elaboration of pb-eb-gall dendrites can be separated into at least two discrete steps: initial growth with R2/R4m axons shortly after the onset of metamorphosis and a second expansion phase from the peripheral R2/R4m axon regions to the outermost R4d axon and central R3 axon regions of the EB during the second day after pupae formation. This two-step dendrite growth strategy by pb-eb-gall neurons may direct R axon lamina/ring formation since R axon lamination is greatly compromised following pb-eb-gall neuron ablation. Selective association between pb-eb-gall dendrites and early extending R2/R4m axons could serve to localize R2/R4m axons in the peripheral EB and constrain late R3/R4d axon growth in the central and outermost EB through short-range attraction and repulsion. Future studies will determine the underlying molecular mechanisms that regulate pb-eb-gall dendrite/R axon local interactions.

R neuron axons also influence pb-eb-gall dendrite development. When we ablated R2/R4m axons, pb-eb-gall neurons show delayed dendrite development at early times and developed smaller dendritic fields within the EB at later stages. However, ablation of either R2/R4m or R1/R3/R4d neurons did not change the general innervation pattern of pb-eb-gall dendrites in the EB, indicating that different R types are independently required for pb-eb-gall dendrite development. In the mouse

cerebellum, the expansion of Purkinjie cell dendrites is regulated by TrkC signaling initiated by granule cell-derived neurotrophin-3 (*Joo et al., 2014*). At present, it is unclear whether R axons merely serve as a substrate for pb-eb-gall dendrite growth or whether they play an active role in pb-eb-gall dendrite expansion.

## Refinement of R neuron laminar organization within the ellipsoid body

Though pb-eb-gall dendrites appear to form a scaffold upon which different R axons extend, interactions among R axons further enhance their laminar organization. Ablation of R2/R4m neurons and their axons, which extend early during EB development, leads to ectopic expansion of R3 axon arbors. Knocking down Sema-1a or PlexA in R2/R4m neurons results in expansion of both R2/R4m and R3 axon rings. These results show that R2/R4m axons constrain R3 axons and separate different R neuron axon layers. However, Sema-1 or PlexA knockdown in R3 neurons does not affect R neuron axon lamination, suggesting that other unidentified molecules mediate direct interactions between R2/R4m and R3 axons, or that R2/R4m neurons regulate R3 axons indirectly through other cells, including pb-eb-gall dendrites.

Within the R2/R4m ring, MARCM analysis shows that Sema-1a in R4m neurons, but not in R2 neurons, cell-autonomously regulates axon morphogenesis and synapse localization. Further, knocking down PlexA, but not Sema-1a, in R2 (along with R1 and R3) resulted in severe R axon lamination defects similar to those observed following expression of *Sema-1a-RNAi* or *PlexA-RNAi* in R2/R4m. Thus, Sema-1a in R4m and PlexA in R2 could act together to modulate R4m and R2 axon branch growth and targeting.

## Versatile roles for conserved guidance cues in EB neuron development

In *Drosophila*, Sema-1a was initially identified as a repulsive axon guidance cue that signals through its receptor PlexA (*Yu et al., 1998*; *Winberg et al., 1998*). More recent studies revealed that Sema-1a also functions as a transmembrane receptor to mediate both axon repulsion and attraction (*Pecot et al., 2013*; *Hsieh et al., 2014*; *Komiyama et al., 2007*; *Lattemann et al., 2007*; *Yu et al., 2010*; *Hernandez-Fleming et al., 2017*; *Cafferty et al., 2006*; *Jeong et al., 2012*). In the mouse visual system, Sema6A is expressed in a group of On direction-selective ganglion cells (On DSGC) and acts as the receptor for brain-derived PlexA2 and PlexA4 to mediate target recognition of On DSGC axons (*Sun et al., 2015*). This suggests that transmembrane semaphorin reverse signaling is conserved across phyla (*Battistini and Tamagnone, 2016*). In the *Drosophila* ellipsoid body, Sema-1a is broadly expressed in most, if not all, major R neuron types. Single-cell MARCM analysis shows that Sema-1a is cell-autonomously required for multiple R neuron types to direct their axon growth within correct EB rings. In addition, axon patterning phenotypes in $Sema\text{-}1a^{-/-}$ mutants are rescued in R4m neurons when full length, but not truncated, Sema-1a is re-introduced into $Sema\text{-}1a$ mutant R neurons, further supporting the idea that Sema-1a reverse signaling controls R axon growth.

However, loss of Sema-1a in different R neuron types and in pb-eb-gall neurons has distinct consequences. In R2/R4m neurons, loss of Sema-1a results in disruption of lamination, whereas in R3 neurons it leads to ectopic axon projections to other brain regions. In pb-eb-gall neurons, both Sema-1a and PlexA act cell-autonomously to guide axon projections outside of the EB but they are not required for dendrite development within the EB. Thus, how Sema-1a forward and reverse signaling shape axon growth and targeting is context dependent. Other parameters, including developmental timing and spatial localization of R neuron axons, also are important for establishing EB ring neuron axon organization. For example, R3 axons develop later than R2/R4m axons and they do not make close contacts with pb-eb-gall dendrites until the second day after puparium formation. Thus, a developmental time window may exist when R3 axons rely on EB-extrinsic repulsive guidance cues, such as fan-shaped body or mushroom body PlexA, to constrain their axon growth through Sema-1a signaling. In addition, other axon guidance and adhesion molecules likely act redundantly or collaboratively with Sema-1a to regulate R axon organization, since we observed that R axon innervation was only partially disrupted in *Sema-1a* and *PlexA* LOF mutants. We expect that many additional molecules contribute to EB ring organization and to the development of other CX structures. Our initial observations on Sema-1a- and PlexA-dependent and independent functions within the developing EB will facilitate their identification. For example, one secreted semaphorin, Sema-2b, and its PlexB receptor exhibit complementary expression patterns in the CX (data not shown).

Our preliminary data suggest that Sema-2b/PlexA signaling also plays important roles in EB morphogenesis and FB lamination.

## Importance of laminated R neuron axons in EB inhibitory circuit organization

Is neuronal lamination essential for synapse organization or a consequence of developmental programs that produce nervous system wiring? In the mouse retina, heterophilic adhesive interactions between bipolar cell axons and their layer-specific targets in the inner plexiform layer are critical for correct lamination and direction selective visual system responses (*Duan et al., 2014*). However, although establishment of strong synaptic specificity between W3B RGCs and VGT3 amacrine cells (ACs) is dependent upon the homophilic Ig superfamily adhesion molecule sidekidk2, loss of sidekick2 does not result in apparent loss of W3B RGC/VGT3 laminar organization (*Krishnaswamy et al., 2015*). Further, in zebrafish *astray* mutants, RGC axons do not show clear lamination in the tectum, yet they do display visual system directional tuning responses similar to wild-type (*Nikolaou and Meyer, 2015*), and in *reeler* mutant mice sensory maps in barrel cortex are normal despite disorganization of cortical laminar patterning (*Guy et al., 2015*). These and additional results (*Zhang et al., 2017*) suggest that lamination may not always be required for functional circuit assembly.

Our results on the role played by Sema-1a in EB axon targeting show that disruption of R axon lamination does indeed result in ectopic inhibitory synaptic connections between R2/R4m and R3 neurons, arguing that lamination is required for the organization of inhibitory synapses among R axons in adjacent EB laminae/rings. How does R axon inhibition and laminar organization of R axon inhibitory synapses contribute to the brain functions and behaviors known to involve R neurons and other EB neurons? Previous studies show that R neurons are functionally heterogeneous, mediating behaviors that include learning and memory involving visual and olfactory sensory responses (*Neuser et al., 2008*; *Ofstad et al., 2011*; *Pan et al., 2009*; *Wang et al., 2008*; *Zhang et al., 2013*) and homeostatic regulation of hunger and sleep (*Dus et al., 2013*; *Liu et al., 2016*). R neuron functions can differ from ring to ring, as exemplified by R3 neurons, but not R4 neurons, being required for visual place learning (*Ofstad et al., 2011*). Even within each ring, R neurons can serve different functions. For example, only a subset of R4m neurons regulate hunger sensation (*Dus et al., 2013*), and a small number of R2 neurons encode sleep drive (*Liu et al., 2016*). Therefore, among co-stratified R neuron axons residing in the same ring, lateral inhibition from one neuron type to another may help fine tune circuit output, and mutual inhibition may serve to shift bipartite behavioral choices from one choice to another: for example, increasing locomotion to forage, or decreasing locomotion to rest. Physical separation of R neuron axons into different rings/laminae is one way to ensure that GABA-mediated inhibition is effectively transmitted from one R neuron axon to others within the same ring, and to minimize GABA-mediated inhibition to R neuron axons in other rings.

Lamination may also be involved in separating different types of inhibitory synapses. We found that inotropic GABA-A receptors and GABA are homogenously distributed in R axon terminals throughout the EB. However, metabotropic GABA-B receptors (GABA-B-R) are preferentially localized in the peripheral region of the EB (*Kahsai et al., 2012*). This raises questions as to whether and how GABA-B receptor-expressing neurons form close contacts with select types of R neuron axons and also how intra-ring R neuron connectivity mechanisms are influenced by different inhibitory circuits.

In conclusion, we have investigated the spatiotemporal innervation by R neuron axons and pb-eb-gall dendrites during EB formation, and we have uncovered different roles played by these cellular components during the formation of the EB circular lamination pattern. Further, the transmembrane proteins Sema-1a and PlexA play critical roles in the development of R neuron axon patterning in the EB. These results provide evidence for the functional significance of R axon lamination in the context of inhibitory synapse organization within and between EB laminae/rings, suggesting that similar developmental strategies are employed by the EB in flies and in other laminated neural systems in both invertebrates and vertebrates. These include step-wise innervation by pre- and post-synaptic processes and a gradual expansion of laminae during neural development as more neuronal processes undergo directed targeting (*Kolodkin and Hiesinger, 2017*). This multi-step process involving neurite growth, guidance, and targeting uses many phylogenetically conserved guidance cues and receptors, and our analysis of EB lamination sets the stage for studying how highly

organized neuronal structural features, including lamination, facilitate select synapse formation, circuit organization and behavior.

## Materials and methods

### Fly strains and genetics

All GMR GAL4 and lexA lines purchased from the Bloomington Drosophila Stock Center (BDSC) at Indiana University were generated at Janelia farm (*Pfeiffer et al., 2008*; *Jenett et al., 2012*). The following transgenes were used: *UAS-CD8::GFP* (RRID: BDSC_5130) (*Lee and Luo, 1999*), *UAS-mCherry* (a kindly gift from Rui Duan in Elizabeth Chen laboratory), *13xlexAop2-6xmCherry-HA* (RRID: BDSC_52272) (*Shearin et al., 2014*), *3xUAS-IVS-Syt::smGFP-HA* (*Aso et al., 2014*), *UAS-Syt::GFP*, *UAS-DenMark* (RRID: BDSC_33064) (*Nicolai˙ et al., 2010*), *UAS-CD4::spGFP$_{1-10}$*, *lexAop-CD4::spGFP$_{11}$* (RRID: BDSC_57321) (*Gordon and Scott, 2009*), *UAS-nSyb::spGFP$_{1-10}$*, *lexAop-CD4::spGFP$_{11}$* (RRID: BDSC_64314) (*Macpherson et al., 2015*), *UAS-DTI* (*Lin et al., 2015*), *tub-GAL80$^{ts}$* (RRID: BDSC_7108), *UAS-Dicer2* (RRID: BDSC_24644) (*Dietzl et al., 2007*), *8xlexAop2-IVS-GAL80* (RRID: BDSC_32215), *13xlexAop2-IVS-CsChrimson::mVenus* (RRID: BDSC_55138) (*Klapoetke et al., 2014*), *UAS-Sema-1a* (RRID: BDSC_65734) (*Jeong et al., 2012*), *UAS-Sema-1a.mEC-5xmyc* (renamed *UAS-Sema-1a$^{EcTM}$* in this paper, RRID: BDSC_65739) (*Jeong et al. (2012)*. The following RNAi lines were ordered from Vienna Drosophila Resource Center (VDRC) (*Dietzl et al., 2007*): *UAS-Sema-1a-RNAi* (GD36148 and KK104505), *UAS-PlexA-RNAi* (GD27238), *UAS-Gad1-RNAi* (GD32344). Additional RNAi lines were ordered from BDSC (*Perkins et al., 2015*): *UAS-Sema-1a-RNAi* (HMS01307, RRID: BDSC_34320), *UAS-PlexA-RNAi* (HM05221, RRID_ BDSC_30483), *UAS-Rdl-RNAi(8–10 j)* (*Liu et al., 2007*). The following mutant alleles were used for gene expression analysis or LOF experiments: *Sema-1a$^{FSF}$* (*Pecot et al., 2013*), *Sema-1a$^{P1}$* (RRID: BDSC_11097) (*Yu et al., 1998*), *Plex-A$^{MB09499}$* (RRID: BDSC_61741) (*Jeong et al., 2012*), *Gad1$^{MI09277}$-QF2* (RRID: BDSC_60323) (*Diao et al., 2015*), *Rdl$^{MI02957}$-GAL4* (RRID: BDSC_60328) (*Diao et al., 2015*), *ChaT$^{MI04508}$-QF2* (RRID: BDSC_60320) (*Diao et al., 2015*), *vGluT$^{MI04979}$-QF2* (RRID: BDSC_60315) (*Diao et al., 2015*).

### Genetic analyses

Flies were reared at 25°C for general purposes.

For genetic ablation experiments, eggs were laid and maggots were kept at 18°C. In about 8–10 days after egg laying, the late third instar larvae crawled out of the food to prepare for pupariation. These wandering third instar larvae of desired genotypes were collected and transferred to new vials kept at 29°C (0 hr after temperature shift). In 24 or 48 hr after the transferring, pupae in the new vials were dissected and brains were stained for analysis. The pupae at 24 and 48 hr after temperature shift are comparable to normal-raised animals at 24 and 48 hr after pupae formation.

For MARCM analyses, the hsFLP was used to generate mosaic clones as previously described (*Lee and Luo, 1999*) with small modifications. To generate small R neuron clones, middle and late 3$^{rd}$ instar larvae were heat shocked twice, for 60 min at 37°C each time, in two consecutive days. Adult male flies were dissected for immunohistochemistry analysis.

For most of RNAi experiments, parental flies were kept at 25°C to lay eggs. One day after egg laying, larvae were transferred to and raised at 29°C until adult F1 flies were dissected. Except for RNAi experiments using *UAS-Sema-1a-RNAi (HMS01307)*, both parental and F1 animals were kept at 25°C.

### Immunohistochemistry

Fly brains were quickly dissected from pupae in cold PBS or from adult flies in cold PBS with 0.1% Triton X-100 (0.1% PBT), and immediately transferred into fixation buffer (4% paraformaldehyde in 0.1% PBT). Brains were notated in fixation buffer for 20 min at room temperature (RT). After washing with 0.1% PBT, fly brains were incubated with blocking buffer (5% normal goat serum in 0.3% PBT) for 1 hr at RT. Then brains were incubated with primary and secondary antibodies for 2 days at 4°C or 1 day at RT for each antibody. Brains were washed intensively (20 min, 3 times in 0.3% PBT at RT) after the primary and secondary antibody incubation. After final wash, brains were incubated with a drop of Vectashield mounting medium (Vector Laboratories, H-1000) overnight at 4°C. Then brains were loaded onto glass slides (Superfrost Plus, Fisherbarnd) prepared with silicon spacers of 120 μm

depth (Grace Biolabs), covered by glass coverslip (1 oz., Fisherbrand) and were ready for imaging analysis.

The following primary antibodies were used: chicken-anti-GFP (1:1000, AVES, RRID: AB_10000240), rabbit-anti-GFP (1:1000, Thermofisher, RRID: AB_221569), mouse-anti-GFP (1:100, Sigma G6539, RRID: AB_259941) (good for GRASP), rabbit-anti-DsRed (1:1000,Clontech, RRID: AB_10013483), rabbit-anti-Sema-1a (1:200) (*Yu et al., 1998*), rabbit-anti-PlexA (1;200, RRID: AB_2569773) (*Sweeney et al., 2007*), rat-anti-HA (1:500, Roche, 3F10, RRID: AB_390915), mouse-anti-Brp (1:50, Developmental Studies Hybridoma Bank (DSHB), Nc82, RRID: AB_2314868), rat-anti-CadN (1:50, DSHB, DN-Ex#8, RRID: AB_2619582), rabbit-anti-GABA (1:500, Sigma, A2052, RRID: AB_477652), rabbit-anti-RDL (1:100, RRID: AB_2568660) (*Liu et al., 2007*), mouse-anti-ChAT (1:100, DSHB 4B1, RRID: AB_528122), rabbit-anti-vGluT (1:5000, RRID: AB_2567386) (*Daniels et al., 2004*), mouse-anti-TH (1:100, EMD Millipore MAB318, RRID: AB_2201528). The secondary antibodies were raised in goat against rabbit, chicken, mouse and rat antisera (Life Technology), conjugated to Alexa 488 (1:1000), Alexa 555 (1:1000) or Alexa 647 (1:300). Antibodies are prepared in the blocking buffer with 0.02% $NaN_3$ and primary antibodies can be reused for several times.

## Image acquisition and processing

All images were taken on a LSM700 confocal microscope (Zeiss) using either a 20X air lens (N.A. 0.8) or a 63X oil immersion lens (N.A. 1.4). Most of the image stacks were taken under 1X zoom, in a 512 × 512 configuration, and have 1 μm (20X lens) and 0.5 μm (63X lens) Z resolution. Unless specified, images stacks were processed with Fiji (imagej) and Adobe Photoshop CS6. Figures are composed with Adobe Illustrator CS6.

## R4m MARCM clone synapse and morphology quantification

Image stacks with CD8-GFP-labeled R4m axon arbors were first reconstructed into three dimension (3D) images in imaris 7.7.3 (Bitplane). The experimenter was blinded to the genotype and did the following analyses. Axonal branches were semi-automatically traced (*Figure 4—figure supplement 1Cii and 1Cvi*, yellow lines) using 'Filament' functions. Pre-synaptic boutons were manually defined based on their enlarged morphologies (*Figure 4—figure supplement 1Ciii and 1Cvii*, white dots). At last, the center of EB canal was semi-manually determined based on Nc82 staining (*Figure 4—figure supplement 1Civ*, white cord) using 'Filament' function. The minimal distance between each bouton to the EB center cord was automatically measured and transformed into voxel intensity using 'Matlab extension', and each bouton is differentially color-coded based on their distance to the EB center (*Figure 4figure supplement 1Civ and 1Cviii*, colorful dots).

## Electrophysiological recordings

Experiments were performed on 3- to 6-day-old female flies, with the experimenter blinded to the genotype. Perforated patch-clamp recordings with $\beta$-escin were performed as previously described with minor modifications (*Liu et al., 2016*), in order to measure action potentials (APs) from EB R2/R4m neurons. Brains were removed and dissected in a *Drosophila* physiological saline solution (101 mM NaCl, 3 mM KCl, 1 mM $CaCl_2$, 4 mM $MgCl_2$, 1.25 mM $NaH_2PO_4$, 20.7 mM $NaHCO_3$, and 5 mM glucose; pH 7.2), which was pre-bubbled with 95% $O_2$ and 5% $CO_2$. To better visualize the recording site, the perineuronal sheath surrounding the brain was focally and carefully removed after treating with an enzymatic cocktail, collagenase (0.4 mg/ml) and dispase (0.8 mg/ml), at 22°C for 1 min and cleaning with a small stream of saline pressure-ejected from a large diameter pipette using a 1-mL syringe. In addition, prior to recording, cell surfaces were cleaned with saline pressure-ejected from a small diameter pipette, using a 1-ml syringe connected to the pipette holder. The recording chamber was placed on an X-Y stage platform (PP-3185–00; Scientifica, UK), and the cell bodies of the targeted EB ring neurons were visualized with tdTomato fluorescence on a fixed-stage upright microscope (BX51WI; Olympus, Japan) and viewed with a 40× water immersion objective lens (LUM-PlanFl, NA: 0.8, Olympus). Patch pipettes (8–12 MΩ) were fashioned from borosilicate glass capillary without filament (OD/ID: 1.2/0.68 mm, 627500, A-M systems, WA) by using a Flaming-Brown puller (P-1000; Sutter Instrument), and further polished with a MF200 microforge (WPI) prior to filling internal pipette solution (102 mM potassium gluconate, 0.085 mM CaCl2, 0.94 mM EGTA, 8.5 mM HEPES, 4 mM Mg-ATP, 0.5 mM Na-GTP, 17 mM NaCl; pH 7.2). Biocytin hydrazide (13 mM; Life

Technologies) was added to the pipette solution before the recording. Recordings were acquired with an Axopatch 200B amplifier (Molecular Devices), and sampled with a Digidata 1440A interface (Molecular Devices). These devices were controlled via pCLAMP 10 software (Molecular Devices). The signals were sampled at 20 kHz and low-pass filtered at 2 kHz. Junction potentials were nullified prior to high-resistance (GΩ) seal formation. Cells showing evidence of 'mechanical' breakthrough, as assessed by the abrupt generation of a large capacitance transient (as opposed to the more progressive, gradual one generated by chemical perforation) were excluded. One neuron per brain was recorded. During the recording, the bath solution was slowly but continuously perfused with saline by means of a gravity-driven system (approximate flow rate of 1–2 ml/min). APs were elicited in response to current injections with 300 ms stepping pulses at 20 pA increments up to 100 pA. Electrophysiological analysis was performed using custom MATLAB-based software (software code is included in *Source code 1*). APs were detected automatically by identification of local maxima and were then manually curated to remove excitatory post-synaptic potentials using minimum voltage threshold criteria. Frequency of detected APs was quantified as mean firing rate during current injection. Several key parameters were calculated or measured from the original data, such as: the slope of the f-I curve, the current threshold and and input resistance and resting membrane potential.

## Optogenetic photostimulation

All-trans retinal (ATR) (R2500, Sigma) was prepared as a 35 mM stock solution dissolved in ethanol, and this stock was mixed into rehydrated fly food flakes (Nutri-Fly Instant, 66–117, Genesee Scientific) at a final concentration of 400 µM. A high-powered red LED with peak wavelength at 627 nm (LXM2-PD01-0050, Lumileds) and Buckpuck driver (RapidLED, Randolph, Vermont) was used to stimulate CsChrimson-expressing EB R3 neurons. The LED was mounted directly underneath the preparation and light was presented at 0.276 mW/mm$^2$ as measured by light meter (Fieldmate power meter, Coherent, CA). Photostimulation began 300 ms before the onset of current injection and ended 300 ms after termination of the current step, which also lasted 300 ms. The timing of photostimulation from LED and current injection from electrode were synchronized using an Arduino Uno board.

## Single-cell labeling

After recording the physiological responses of EB ring neurons, biocytin hydrazide was iontophoresed into the cell with a constant hyperpolarizing current of 0.9–1.2 nA passed for at least 5 min. The brain was then fixed in 4% paraformaldehyde in PBS overnight at 4°C. After washing for 1 hr in several changes of PBST (0.3% Triton X-100 in PBS) at room temperature, brains were incubated with Alexa-647 conjugated Streptavidin (1:300, Molecular Probes, ThermoFisher Sci.) in PBST overnight at 4°C. After extensive washing (15 min, 3 times in PBST), brains were processed following standard fluorescent immunostaining protocol.

## Statistical analysis

All statistical tests were performed by Prism 6 or 7 (GraphPad); methods are listed below. Post hoc power analysis was conducted on existing datasets by G*Power 3.1 (Universität Düsseldorf). All tests that have p<0.05 also have power >0.8 except *Figure 7—figure supplement 1C*, which has a power value of 0.728.

| Figures | Statistic methods |
|---|---|
| *Figure 2, E*;<br>*Figure 2—figure supplement 1, D and E*;<br>*Figure 6, C and F* | Two-tailed unpaired t-test with Mann-Whitney test.<br>''<br>'' |
| *Figure 2, D*;<br>*Figure 5—figure supplement 1, G* | Multiple t-tests with the Holm-Sidak correction.<br>'' |
| *Figure 4, B,F–H*;<br>*Figure 4—figure supplement 1, D–F*;<br>*Figure 5, B–D*;<br>*Figure 5—figure supplement 1, D* | One-way ANOVA with multiple comparisons (Tukey test)<br>''<br>''<br>'' |

Figure 7E and H;
Two-tailed paired t-tests
Figure 7—figure supplement 1, A–D
''

## Genotypes in figures
### Figure 1
*w; R32H08-lexA, lexAop-mCherry; R15F02-GAL4, UAS-mCD8::GFP*

### Figure 1-Figure supplement 1
#### Panels A, C and E
*hsFLP, UAS-mCD8::GFP/Y; FRT40A/tub-GAL80, FRT40A; R15B07-GAL4/+*

#### Panels B and D
*hsFLP, UAS-mCD8::GFP/Y; FRT40A/tub-GAL80, FRT40A; R20D01-GAL4/+*

#### Panel F
*hsFLP, UAS-mCD8::GFP/Y; FRT40A/tub-GAL80, FRT40A; R19G02-GAL4/+*

#### Panel H
*w; R19G02-GAL4/UAS-Syt::GFP, UAS-DenMark*

#### Panel I
*w; R19G02-lexA/+; R15B07-GAL4/UAS-CD4::spGFP1-10, lexAop-CD4::spGFP11*

#### Panel J
*w; R19G02-lexA/+; R20D01-GAL4/UAS-CD4::spGFP1-10, lexAop-CD4::spGFP11*

#### Panel K
*w; R32H08-lexA, lexAop-mCherry; R19G02-GAL4, UAS-mCD8::GFP*

#### Panel L
*w; R19G02-lexA, lexAop-mtdT; R15F02-GAL4, UAS-mCD8::GFP*

### Figure 2
#### Panel B
*w; R32H08-lexA, lexAop-mCherry/tub-GAL80$^{ts}$; R19G02-GAL4, UAS-mCD8::GFP/+*
*    w; R32H08-lexA, lexAop-mCherry/tub-GAL80$^{ts}$, UAS-DTI; R19G02-GAL4, UAS-mCD8::GFP/UAS-DTI*

#### Panels C-E
*w; R54B05-lexA, lexAop-mtdT/tub-GAL80$^{ts}$; R32H08-GAL4, UAS-mCD8::GFP/+*
*    w; R54B05-lexA, lexAop-mtdT/tub-GAL80$^{ts}$, UAS-DTI; R32H08-GAL4, UAS-mCD8::GFP/UAS-DTI*

### Figure 2-Figure supplement 1
#### Panel A
*w; R32H08-lexA, lexAop-mCherry/tub-GAL80$^{ts}$; R19G02-GAL4, UAS-mCD8::GFP/+*
*    w; R32H08-lexA, lexAop-mCherry/tub-GAL80$^{ts}$, UAS-DTI; R19G02-GAL4, UAS-mCD8::GFP/UAS-DTI*

#### Panel B
*w; R70B04-lexA, lexAop-mCherry/tub-GAL80$^{ts}$; R19G02-GAL4, UAS-mCD8::GFP/+*
*    w; R70B04-lexA, lexAop-mCherry/tub-GAL80$^{ts}$, UAS-DTI; R19G02-GAL4, UAS-mCD8::GFP/UAS-DTI*

Panels C–E

*w; R19G02-lexA, lexAop-mCherry/tub-GAL80^ts; R32H08-GAL4, UAS-mCD8::GFP/+*
   *w; R19G02-lexA, lexAop-mCherry/tub-GAL80^ts, UAS-DTI; R32H08-GAL4, UAS-mCD8::GFP/UAS-DTI*

Panel F

*w; R54B05-lexA, lexAop-mtdT/tub-GAL80^ts; R32H08-GAL4, UAS-mCD8::GFP/+*
   *w; R54B05-lexA, lexAop-mtdT/tub-GAL80^ts, UAS-DTI; R32H08-GAL4, UAS-mCD8::GFP/UAS-DTI*

## Figure 3
Panels A and B
*w^1118*

Panel D (left)

*w; Sema-1a-FSF/+; UAS-FLP, LexAop-CD2::GFP/+*

Panels D(right), E and F

*w; Sema-1a-FSF/+; UAS-FLP, LexAop-CD2::GFP/R11F03-GAL4*

Panel H

*w; Sema-1a-FSF/+; UAS-FLP, LexAop-mCherry/R11F03-GAL4, UAS-mCD8::GFP*

Panel I

*w; Sema-1a-FSF/+; UAS-FLP, LexAop-mCherry/R11F03-GAL4, UAS-mCD8::GFP*

Panel J

*w; Sema-1a-FSF/+; UAS-FLP, LexAop-mCherry/EB1-GAL4, UAS-mCD8::GFP*

## Figure 3-Figure supplement 1
Panels A and B
*w^1118*

## Figure 4
Panels A and B

*UAS-Dicer2/+; R32H08-GAL4, UAS-Syt::GFP/+*
   *UAS-Dicer2/+; R32H08-GAL4, UAS-Syt::GFP/UAS-Sema-1a-RNAi(GD36148)*
   *UAS-Dicer2/+; UAS-Sema-1a-RNAi(KK104505)/+; R32H08-GAL4, UAS-Syt::GFP/+*
   *UAS-Dicer2/+; R32H08-GAL4, UAS-Syt::GFP/UAS-PlexA-RNAi(GD27238)*
   *UAS-Dicer2/+; R32H08-GAL4, UAS-Syt::GFP/UAS-PlexA-RNAi(HM05211)*

Panel C

**Control**: *hsFLP, UAS-mCD8::GFP/Y; FRT40A/tub-GAL80, FRT40A; VT31157-GAL4/UAS-Syt-HA*
   **Sema-1a^-/-**: *hsFLP, UAS-mCD8::GFP/Y; Sema-1a^P1, FRT40A/tub-GAL80, FRT40A; VT31157-GAL4/UAS-Syt-HA*

Panels D, F–H

**Control**: *hsFLP, UAS-mCD8::GFP/Y; FRT40A/tub-GAL80, FRT40A; R20D01-GAL4/UAS-Syt-HA*
   **Sema-1a^-/-**: *hsFLP, UAS-mCD8::GFP/Y; Sema-1a^P1, FRT40A/tub-GAL80, FRT40A; R20D01-GAL4/UAS-Syt-H*
   **Sema-1a rescue**: *hsFLP, UAS-mCD8::GFP/Y; Sema-1a^P1, FRT40A/tub-GAL80, FRT40A; R20D01-GAL4/UAS-Sema-1a*
   **Sema-1a^EcTM rescue**: *hsFLP, UAS-mCD8::GFP/Y; Sema-1a^P1, FRT40A/tub-GAL80, FRT40A; R20D01-GAL4/UAS-Sema-1a^EcTM*

## Figure 4-Figure supplement 1
### Panels A and B
**Control**: *hsFLP, UAS-mCD8::GFP/Y; FRT40A/tub-GAL80, FRT40A; VT31157-GAL4/UAS-Syt-HA*
   **Sema-1a$^{-/-}$**: *hsFLP, UAS-mCD8::GFP/Y; Sema-1a$^{P1}$, FRT40A/tub-GAL80, FRT40A; VT31157-GAL4/ UAS-Syt-HA*

## Figure 4-Figure supplement 2
### Panel A
**Control**: *UAS-Dicer2/+; R34D03-lexA, lexAop-mtdT/+; R40G10-GAL4, UAS-mCD8::GFP/+*
   **Sema-1a-RNAi:** *UAS-Dicer2/+; R34D03-lexA, lexAop-mtdT/ UAS-Sema-1a-RNAi(KK104505); R40G10-GAL4, UAS-mCD8::GFP/+* and
   *UAS-Dicer2/+; R34D03-lexA, lexAop-mtdT/+; R40G10-GAL4, UAS-mCD8::GFP/ UAS-Sema-1a-RNAi(GD36148)*
   **PlexA-RNAi:** *UAS-Dicer2/+; R34D03-lexA, lexAop-mtdT/+; R40G10-GAL4, UAS-mCD8::GFP/ PlexA-RNAi (GD27238)* and
   *UAS-Dicer2/+; R34D03-lexA, lexAop-mtdT/+; R40G10-GAL4, UAS-mCD8::GFP/ UAS-PlexA-RNAi (HM05211)*

### Panel B
**Control**: *UAS-Dicer2/+; R54D05-lexA, lexAop-mtdT/+; R40G10-GAL4, UAS-mCD8::GFP/+*
   **Sema-1a-RNAi:** *UAS-Dicer2/+; R54D05-lexA, lexAop-mtdT/ UAS-Sema-1a-RNAi(KK104505); R40G10-GAL4, UAS-mCD8::GFP/+* and
   *UAS-Dicer2/+; R54D05-lexA, lexAop-mtdT/+; R40G10-GAL4, UAS-mCD8::GFP/ UAS-Sema-1a-RNAi(GD36148)*
   **PlexA-RNAi:** *UAS-Dicer2/+; R54D05-lexA, lexAop-mtdT/+; R40G10-GAL4, UAS-mCD8::GFP/ PlexA-RNAi (GD27238)* and
   *UAS-Dicer2/+; R54D05-lexA, lexAop-mtdT/+; R40G10-GAL4, UAS-mCD8::GFP/ UAS-PlexA-RNAi (HM05211)*

## Figure 5
### Panels A-D
**Control**: *UAS-Dicer2/+; R54B05-lexA, lexAop-mtdT/+; R32H08-GAL4, UAS-mCD8::GFP/+*
   **Sema-1a-RNAi(KK):** *UAS-Dicer2/+; R54B05-lexA, lexAop-mtdT/ UAS-Sema-1a-RNAi(KK104505); R32H08-GAL4, UAS-mCD8::GFP/+*
   **Sema-1a-RNAi(GD):** *UAS-Dicer2/+; R54B05-lexA, lexAop-mtdT/+; R32H08-GAL4, UAS-mCD8:: GFP/UAS-Sema-1a-RNAi(GD36148)*
   **PlexA-RNAi(GD):** *UAS-Dicer2/+; R54B05-lexA, lexAop-mtdT/+;R32H08-GAL4, UAS-mCD8::GFP/ UAS-PlexA-RNAi(GD27238)*
   **PlexA-RNAi(HM):** *UAS-Dicer2/+; R54B05-lexA, lexAop-mtdT/+;R32H08-GAL4, UAS-mCD8::GFP/ UAS-PlexA-RNAi(HM05211)*

## Figure 5-Figure supplement 1
### Panel A
*w;; R54B05-lexA, lexAop-mtdT/+; R32H08-GAL4, UAS-mCD8::GFP/+*

### Panel B
*w;; R54B05-lexA, lexAop-mtdT/+; R32H08-GAL4, UAS-mCD8::GFP/UAS-Sema-1a-RNAi(HMS01307)*

### Panels C and D
**Control**: *UAS-Dicer2/+; R46D01-GAL4, UAS-mCD8::GFP/+*
   **Sema-1a-RNAi(KK):** *UAS-Dicer2/+; UAS-Sema-1a-RNAi(KK104505)/+; R46D01-GAL4, UAS-mCD8::GFP/+*
   **Sema-1a-RNAi(GD):** *UAS-Dicer2/+; R46D01-GAL4, UAS-mCD8::GFP/UAS-Sema-1a-RNAi (GD36148)*
   **PlexA-RNAi(GD):** *UAS-Dicer2/+; R46D01-GAL4, UAS-mCD8::GFP/UAS-PlexA-RNAi(GD27238)*

**PlexA-RNAi(HM):** *UAS-Dicer2/+; R46D01-GAL4, UAS-mCD8::GFP/UAS-PlexA-RNAi(HM05211)*

### Panels F-G
**Control:** *hsFLP, UAS-mCD8::GFP/Y; FRT40A/tub-GAL80, FRT40A; R84H09-GAL4/+*

**Sema-1a$^{-/-}$:** *hsFLP, UAS-mCD8::GFP/Y; Sema-1a$^{P1}$, FRT40A/tub-GAL80, FRT40A; R84H09-GAL4/+*

## Figure 5-Figure supplement 2
### Panel A
**Control:** *UAS-Dicer2/+; R19G02-lexA, lexAop-mtdT/+; R32H08-GAL4, UAS-mCD8::GFP/+*

**Sema-1a-RNAi:** *UAS-Dicer2/+; R19G02-lexA, lexAop-mtdT/ UAS-Sema-1a-RNAi(KK104505); R32H08-GAL4, UAS-mCD8::GFP/+* and

*UAS-Dicer2/+; R19G02-lexA, lexAop-mtdT/+; R32H08-GAL4, UAS-mCD8::GFP/ UAS-Sema-1a-RNAi(GD36148)*

**PlexA-RNAi:** *UAS-Dicer2/+; R19G02-lexA, lexAop-mtdT/+; R32H08-GAL4, UAS-mCD8::GFP/ PlexA-RNAi (GD27238)* and

*UAS-Dicer2/+; R19G02-lexA, lexAop-mtdT/+; R32H08-GAL4, UAS-mCD8::GFP/ UAS-PlexA-RNAi (HM05211)*

### Panel B
**Control:** *w; R19G02-lexA, lexAop-mtdT/UAS-Syb-spGFP$_{1-10}$, lexAop-CD4-spGFP$_{11}$; R32H08-GAL4, lexAop-GAL80/+*

**Sema-1a-RNAi:** *w; R19G02-lexA, lexAop-mtdT/UAS-Syb-spGFP$_{1-10}$, lexAop-CD4-spGFP$_{11}$; R32H08-GAL4, lexAop-GAL80/UAS-Sema-1a-RNAi(HMS01307)*

### Panels C and D
**Control:** *UAS-Dicer2/+; R32H08-lexA (or R70B04-lexA), lexAop-mtdT/+; R19G02-GAL4, UAS-mCD8::GFP/+*

**Sema-1a-RNAi:** *UAS-Dicer2/+; R32H08-lexA (or R70B04-lexA), lexAop-mtdT/ UAS-Sema-1a-RNAi (KK104505); R19G02-GAL4, UAS-mCD8::GFP/+* and

*UAS-Dicer2/+; R32H08-lexA (or R70B04-lexA), lexAop-mtdT/+; R19G02-GAL4, UAS-mCD8::GFP/ UAS-Sema-1a-RNAi(GD36148)*

**PlexA-RNAi:** *UAS-Dicer2/+; R32H08-lexA (or R70B04-lexA), lexAop-mtdT/+; R19G02-GAL4, UAS-mCD8::GFP/PlexA-RNAi (GD27238)* and

*UAS-Dicer2/+; R32H08-lexA (or R70B04-lexA), lexAop-mtdT/+; R19G02-GAL4, UAS-mCD8::GFP/ UAS-PlexA-RNAi(HM05211)*

## Figure 6
### Panel A
*w; Gad1$^{MI09277}$-QF2/QUAS-mtdT*

### Panels B-C
*UAS-Dicer2/+; R32H08-GAL4, UAS-CD8::GFP/+*

*UAS-Dicer2/+; R32H08-GAL4, UAS-CD8::GFP/UAS-Gad1-RNAi(GD32344)*

### Panel D
*w; Rdl1$^{MI029577}$-GAL4/UAS-mCD8::GFP*

### Panels E-F
*UAS-Dicer2/+; R32H08-GAL4, UAS-CD8::GFP/+*

*UAS-Dicer2/+; R32H08-GAL4, UAS-CD8::GFP/UAS-Rdl-RNAi(8–10 j)*

## Figure 6-Figure supplement 1
### Panels A, C and E
*w$^{1118}$*

Panel B
*w; ChaT$^{MI045087}$-QF2/QUAS-mtdT*

Panel D
*w; vGluT$^{MI04979}$-GAL4/+; QUAS-mtdT /+*

Panel F
*w; TH-GAL4/UAS-mCD8::GFP*

## Figure 7 and Figure 7-Figure supplement 1

Control
*w; R54B05-lexA, lexAop2-GAL80/lexAop2-CsChrimson-mVenus; R32H08-GAL4, UAS-mCherry/+*

Sema-1a-RNAi (R2/R4m)
*w; R54B05-lexA,lexAop2-GAL80/lexAop2-CsChrimson-mVenus; R32H08-GAL4, UAS-mCherry/UAS-Sema-1a-RNAi(HMS01307)*

## Acknowledgements

We are very grateful to Christopher Potter for helpful comments on the manuscript and discussions, members of the Kolodkin, Wu, and Potter laboratories for helpful discussions throughout the course of this project, and V Hartenstein for communication of results prior to publication. We thank R Davis, A DiAntonio, and L Luo for antibodies; E Chen, M Reiser, S L Zipursky, The Bloomington Stock Center, and The Vienna Drosophila Resource Center for fly stocks; and M Pucak and the NINDS Multi-photon Imaging Core Facility at JHMI (P30 NS050274) for imaging and data analysis. This work was supported by grants from the NIH (R01NS079584 to MNW), the Japan Society for the Promotion of Science (MT), and the Howard Hughes Medical Institute (XX, MPB, SPM and ALK). ALK is an Investigator of the Howard Hughes Medical Institute.

## Additional information

### Funding

| Funder | Grant reference number | Author |
| --- | --- | --- |
| Howard Hughes Medical Institute | | Xiaojun Xie<br>Matthew P Brown<br>Sarah P Mitchell<br>Alex L Kolodkin |
| National Institutes of Health | 1R01 NS079584 | Xiaojun Xie<br>Matthew P Brown<br>Sarah P Mitchell<br>Alex L Kolodkin |
| National Institutes of Health | P30 NS50274 | Xiaojun Xie<br>Matthew P Brown<br>Sarah P Mitchell<br>Alex L Kolodkin |
| National Institutes of Health | 1R21 NS088521 | Masashi Tabuchi<br>Mark N Wu |

The funders had no role in study design, data collection and interpretation, or the decision to submit the work for publication.

### Author contributions

XX, Conceptualization, Resources, Data curation, Formal analysis, Validation, Visualization, Methodology, Writing—original draft, Project administration, Writing—review and editing; MT, Conceptualization, Data curation, Formal analysis, Methodology, Writing—review and editing; MPB, Data curation, Formal analysis, Methodology; SPM, Conceptualization, Resources, Data curation; MNW, Conceptualization, Supervision, Funding acquisition, Project administration, Writing—review and

editing; ALK, Conceptualization, Data curation, Supervision, Funding acquisition, Validation, Investigation, Methodology, Project administration, Writing—review and editing

## Author ORCIDs

Xiaojun Xie, http://orcid.org/0000-0003-3459-6095
Alex L Kolodkin, http://orcid.org/0000-0001-7562-5513

## Additional files

**Supplementary files**
• Source code 1. Electrophysiology MATLAB code.

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
