## [Decision Letter]

Thank you for submitting your article "Semaphorin-mediated Lamination Facilitates Inhibitory Synapse Organization and Function in the *Drosophila* Ellipsoid Body" for consideration by *eLife*. Your article has been reviewed by two peer reviewers, and the evaluation has been overseen by Mani Ramaswami as Reviewing Editor and K VijayRaghavan as the Senior Editor. The following individual involved in review of your submission has agreed to reveal his identity: Greg J Bashaw (Reviewer #1).

The reviewers have discussed the reviews with one another and the Reviewing Editor has drafted this decision to help you prepare a revised submission.

Summary:

In this paper Kolodkin and co-workers explore the role of Semaphorin 1a (Sema 1a) and Plexin A in regulated the formation of the circuitry in the ellipsoid body (EB) in the *Drosophila* central brain. They demonstrate that these well-characterized guidance molecules are necessary for the normal lamination of the ellipsoid body neuropil. Using electrophysiology and optogenetics, they demonstrate further that as a consequence of disrupted lamination, neurons form inappropriate (inhibitory) connections with non-target subclasses of neurons. These data argue that lamination is essential to promote normal patterns of synaptic connectivity.

Visualizing Ring (R) axon targeting in the EB using specific Gal4 and LexA lines, the authors demonstrate through ablation experiments that axon-axon interactions between R neurons contribute to the laminar organization of the EB. The transmembrane receptor Sema1a and its interacting partner PlexinA are broadly expressed in R neurons, and knockdown experiments demonstrate they are required for constraining R2/4m axons to specific lamina. Single cell MARCM clones reveal that Sema1a acts cell autonomously in R4m neurons to limit axonal growth and synapse formation, in a cytodomain-dependent manner. In addition, loss of Sema1a or PlexA in R2/4m neurons non-cell autonomously disrupts laminar target of R3 neurons. Protein trap reporters and antibodies reveal that R neurons are primarily GABAergic. Optogenetics and electrophysiological recordings show that while under control conditions, activating R3 neurons has no significant effect on evoked action potentials recorded in R2/4m neurons, when lamination decfects are induced by Sema/Plexin disruption, then R3 neurons inhibit R2/4m neurons.

The Sema/Plexin phenotypes are carefully documented and quite convincing. This study is also unique for experimentally demonstrating a role for lamination in restricting inhibitory connections, going beyond previous studies (Duan et al., 2014) and testing the intuitive but unproven idea that lamination plays an important role in preventing the formation of inappropriate synapses.

This is a valuable contribution. However, it could be improved and deepened by some revisions and clarifications that the authors should try to include, unless there are strong reasons why these would be difficult in a two-month timeframe.

A relatively major comment:

1) Though enticing to think PlexA would be required in R2, this is not experimentally addressed and so the developmental mechanism by which Sema1 and PlexA regulate lamination is poorly defined. That is, while both ligand and receptor are required for lamination, it remains unclear whether they are required on the same or different neurons. Can this issue be experimentally addressed?

---

## [Author Response]

1) Though enticing to think PlexA would be required in R2, this is not experimentally addressed and so the developmental mechanism by which Sema1 and PlexA regulate lamination is poorly defined. That is, while both ligand and receptor are required for lamination, it remains unclear whether they are required on the same or different neurons. Can this issue be experimentally addressed?

This is an important point. We were initially unable to directly test this idea because all the R2 drivers we examined also drive expression in R4m neurons. Until recently, we were unaware of the driver R40G10-GAL4, which was characterized by Dr. Volker Hartenstein’s laboratory and will soon be published in The Journal of Comparative Neurobiology. However, based on their findings (communicated to us as a “personal communication” manuscript now “in press”), R40G10-GAL4 drives expression in R1-3 neurons from early pupae to adult stages. Although R40G10-GAL4 is not R2-specific, it does allow us to separate R2 neurons from R4m neurons during pupal development.

Therefore, we knocked down Sema-1a or PlexA using R40G10-GAL4 and found that both of these manipulations gave strong overall EB morphological and also R axon elaboration defects (New Figure 4—figure supplement 2). Since our previous work showed that knocking down Sema-1a or PlexA with several R1/R3 GAL4 drivers (Figure 5—figure supplement 1 and “data not shown”) did not cause obvious EB morphology or R axon lamination defects, these defects seen using R40G10-GAL4 are likely due to loss of Sema-1a or PlexA from R2 alone, or a combinatorial effect resulting from removing Sema-1a or PlexA from R1/2/3.

We observed significant differences between Sema-1a-RNAi and PlexA-RNAi experiments using R40G10-GAL4. First, EB morphology changes were distinct. PlexA-RNAi resulted in a more open EB conformation than we observed with Sema-1a-RNAi (New Figure 4—figure supplement 2: compare Av and Bv, to Avi and Bvi). Second, R axon lamination was disrupted differently when Sema1a or PlexA was knocked down using R40G10-GAL4 to drive RNAi expression. Two different lexA drivers were used to label R neurons: R34D03-LexA for R4m and R54B05-LexA for R3. These were used separately in combination with R40G10-GAL4 to drive expression of UAS-Sema-1a-RNAi or UAS-PlexA-RNAi.

In Sema-1a–RNAi-expressing animals, although both R4m and R3 axons become disorganized, but their axons remained in the outer and inner regions in the EB, respectively, suggesting preservation of axon lamination (New Figure 4—figure supplement 2, Aii and Bii). However, in PlexA-RNAi expressing animals, R4m and R3 axons expand their elaboration to cover most of the EB, as can be appreciated in frontal view images, presumably resulting in a complete breakdown of R axon lamination (New Figure 4—figure supplement 2, Aiii and Biii). These new data suggest that Sema-1a and PlexA function differently in R1-3 for EB development (New Figure 4—figure supplement 2). Most importantly, PlexA, but not Sema-1a, is required in R1-3 for R axon lamination.

Taken together, our new and previous data support a model in which Sema-1a and PlexA function respectively in R4m and R2 to regulate R axon lamination. The new findings are presented in our New Figure 4—figure supplement 2, and are described in the text of our revised submission.